# VersaPRM: Multi-Domain Process Reward Model via Synthetic Reasoning Data

**Thomas Zeng** [1]  **Shuibai Zhang** [1]  **Shutong Wu** [1]  **Christian Classen** [1]  **Daewon Chae** [2]  **Ethan Ewer** [1]  **Minjae Lee** [3]
**Heeju Kim** [3]  **Wonjun Kang** [3]  **Jackson Kunde** [1]  **Ying Fan** [1]  **Jungtaek Kim** [1]  **Hyung Il Koo** [3]
**Kannan Ramchandran** [4]  **Dimitris Papailiopoulos** [1]  **Kangwook Lee** [1]

## Abstract

Process Reward Models (PRMs) have proven effective at enhancing mathematical reasoning for Large Language Models (LLMs) by leveraging increased inference-time computation. However, they are predominantly trained on mathematical data and their generalizability to non-mathematical domains has not been rigorously studied. In response, this work first shows that current PRMs have poor performance in other domains. To address this limitation, we introduce *VersaPRM*, a multi-domain PRM trained on synthetic reasoning data generated using our novel data generation and annotation method. VersaPRM achieves consistent performance gains across diverse domains. For instance, in the MMLU-Pro category of Law, VersaPRM via weighted majority voting, achieves a 7.9% performance gain over the majority voting baseline—surpassing Qwen2.5-Math-PRM's gain of 1.3%. We further contribute to the community by open-sourcing all data, code and models for VersaPRM.

## 1. Introduction

Large Language Models (LLMs) have demonstrated significant potential in tackling complex reasoning tasks. Specifically, they can employ a step-by-step Chain-of-Thought (CoT) approach to generate more accurate and reliable solutions (Wei et al., 2022b; Kojima et al., 2022; Yao et al., 2023; Madaan et al., 2023). Moreover, by using additional test-time computation, the reasoning performance of LLMs can be further enhanced (Snell et al., 2024; Yao et al., 2024).

An important and widely-adopted test-time computation method is using *external verifiers*, such as reward models

[1]University of Wisconsin–Madison [2]Korea University [3]FuriosaAI [4]University of California, Berkeley. Correspondence to: Kangwook Lee <kangwook.lee@wisc.edu>.

*Proceedings of the $42^{nd}$ International Conference on Machine Learning*, Vancouver, Canada. PMLR 267, 2025. Copyright 2025 by the author(s).

to rank multiple generated solutions and select the best answer (Lightman et al., 2024). Reward models evaluate the quality of solutions, helping guide LLMs toward better outputs. In particular, Outcome Reward Models (ORMs) are used to provide supervision based solely on the correctness of the final outcome. However, ORMs fail to address errors in intermediate steps, limiting their effectiveness for complex, multi-step reasoning tasks (Luo et al., 2024; Lightman et al., 2024; Sun et al., 2024). Because ORMs suffer from this limitation, Process Reward Models (PRMs) have been proposed to offer *fine-grained, step-by-step feedback* on the correctness of each reasoning step (Lightman et al., 2024; Uesato et al., 2022). PRMs have proven highly effective during inference, improving the reranking of generated solutions and guiding LLMs through search-based algorithms (Wan et al., 2024; Wang et al., 2024a).

Several studies have shown that PRMs trained on extensive process supervision significantly outperform ORMs in mathematical reasoning tasks, with notable improvements reported on datasets such as MATH500 and GSM800K (Luo et al., 2024; Lightman et al., 2024; Uesato et al., 2022). While substantial investigation has been made in creating training data (Lightman et al., 2024; Wang et al., 2024b), training PRMs (Xiong et al., 2024), and evaluation (Zheng et al., 2024) with respect to mathematical reasoning, the application of PRMs to *non-mathematical domains*—such as Biology, Chemistry, and Law—remains underexplored. To investigate the capability of math PRMs in non-mathematical domains, we test open-source math PRMs such as Math-Shepherd (Wang et al., 2024b) and Qwen-2.5-Math-PRM (Zheng et al., 2024). Not surprisingly, these PRMs demonstrate poor performance, indicating their limited domain generalizability. They exhibit only marginal improvements over the baseline in Law, Philosophy, and Biology as illustrated in Figure 1.

To address this limitation, we propose fine-tuning PRMs on a synthetically generated multi-domain CoT dataset, to significantly enhance reasoning capabilities beyond mathematics. We call this resulting multi-domain PRM *VersaPRM*, short for versatile PRM. Notably, by sampling questions from the MMLU-Pro dataset (Wang et al., 2024c), we generate CoTs to produce step-by-step reasoning using an

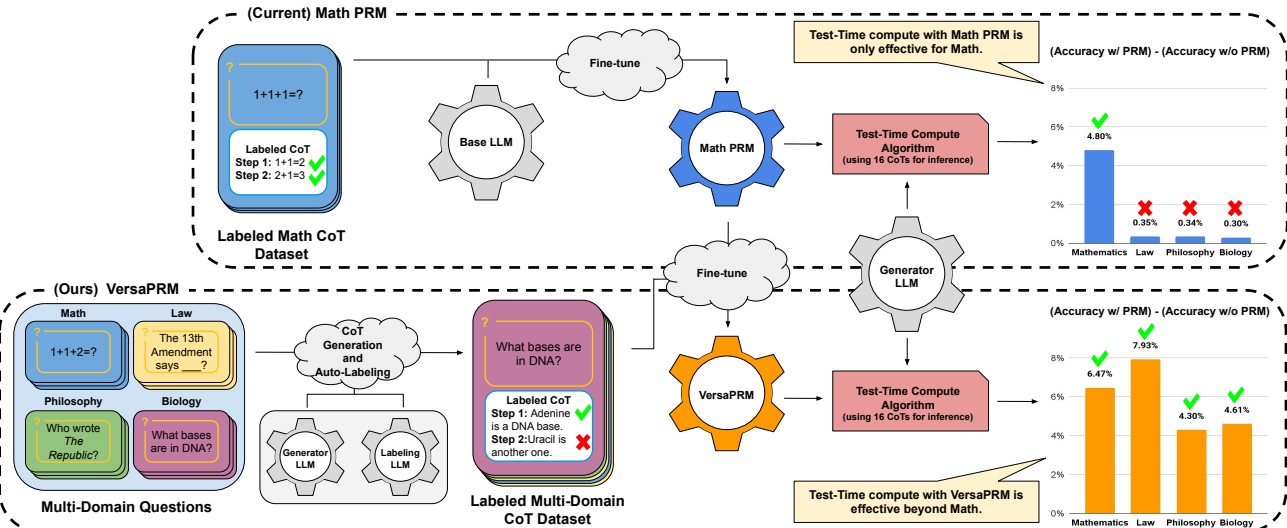

*Figure 1.* Existing open-source PRMs trained on math datasets achieve strong math performance and can outperform a majority voting baseline when used via weighted majority voting. However, these PRMs fail to generalize to other domains (e.g., Law, Philosophy, and Biology), performing no better than the baseline. We propose a multi-domain PRM, VersaPRM, obtained by further fine-tuning a math PRM on a synthetically generated multi-domain dataset. The resulting PRM effectively generalizes beyond math, improving test-time reasoning across multiple domains.

LLM-based generator, i.e., Llama-3.1-8B-Instruct (Dubey et al., 2024), and then auto-label them using an LLM-based labeler, i.e., Llama-3.1-70B-Instruct (Dubey et al., 2024). VersaPRM, which is trained on the resulting synthetic multi-domain reasoning data, shows strong performance across diverse domains. We validate its superior performance using various empirical analyses on VersaPRM against existing open-source PRMs.

Our contributions are summarized as follows:

1. We identify the limited domain generalizability of open-source math PRMs in Section 4.

2. We propose a novel data generation and annotation pipeline across multiple domains in Section 5.

3. We introduce a large-scale, high-quality multi-domain process supervision dataset, dubbed *MMLU-Pro-CoT-Train (Labeled)*.

4. We train a well-generalized PRM that outperforms existing baselines, demonstrating strong generalization across diverse domains in Section 6.

5. We open-source the implementation of VersaPRM with training details, our multi-domain reasoning data, and its model checkpoint; available at `https://github.com/UW-Madison-Lee-Lab/VersaPRM`.

## 2. Related Work

**Chain-of-Thought and Process Reward Models.** PRMs have proven more effective than ORMs in enhancing LLM reasoning via CoT, particularly for mathematical tasks (Luo et al., 2024; Lightman et al., 2024; Sun et al., 2024). Unlike ORMs, which focus on final outcomes, PRMs provide step-by-step feedback, improving error detection in intermediate steps and multi-step tasks (Luo et al., 2024; Lightman et al., 2024; Uesato et al., 2022; Wang et al., 2024b). Techniques like OmegaPRM (Luo et al., 2024) and Math-Shepherd (Wang et al., 2024b) reduce reliance on costly human annotations, while RLHflow (Xiong et al., 2024), OpenR (Wang et al., 2024a) and ProcessBench (Zheng et al., 2024) advance PRM evaluation and training. However, the expertise of existing PRMs is mainly limited to mathematical reasoning. This mathematical focus aligns with findings from Sprague et al. (2025), who show that CoT prompting yields its largest gains on mathematical problems, with only modest improvements in other domains. In contrast, our work demonstrates that when CoT prompting is paired with VersaPRM at test time, consistent improvements can still be achieved across several non-mathematical domains.

**Test-Time Inference Algorithms.** Test-time inference algorithms enhance LLM reasoning by adding computation during inference, offering a distinct axis of scaling compared to the conventional trade-off between training larger models versus smaller, specialized architectures

(Hoffmann et al., 2022). AlphaCode (Li et al., 2022) shows how test-time computing boosts competitive programming performance, while Snell et al. (2024) argue that scaling inference-time compute, rather than model parameters, yields better results by adapting compute allocation to prompt difficulty. Test-time inference includes verifier free methods—like Tree of Thought reasoning (Yao et al., 2024), self-verification (Weng et al., 2023) and stepwise self-evaluation (Xie et al., 2024)—as well as external-verifier methods such as verifier reranking (Cobbe et al., 2021; Wan et al., 2024), tool feedback (Gou et al., 2024), reward-guided reasoning (Yang et al., 2024b), and multi-agent debate (Du et al., 2024). Our work uses PRMs for reranking solutions and guiding reasoning within the external verifier paradigm.

**Synthetic Data Generation.** Obtaining fine-grained step-wise labeling of CoTs via expert annotation is costly and time-consuming. Automated annotation methods, such as rollout-based approaches, reduce human effort but require numerous model inference, which is computationally expensive. Recent effort to mitigate these limitations include multi-rollout generation per reasoning step (Wang et al., 2024b;d) and efficiency improvements via binary search (Luo et al., 2024). Despite these advancements, the reliance on multiple model calls remains a bottleneck. Li et al. (2024) introduce a novel arithmetical-puzzles and show that fine-tuning on large-scale synthetic examples yield precise multi-step math reasoning capabilities—though its applicability outside mathematics remains unclear.

This challenge of efficiently generating supervision data mirrors the goals of knowledge distillation, where insights from a larger "teacher" model are transferred to a smaller "student" model, often by using the teacher to generate labeled training data (Hinton et al., 2015; Gu et al., 2024). Indeed, recent studies by Gilardi et al. (2023) and Fonseca & Cohen (2024) demonstrate the potential of LLMs as data labelers. The effectiveness of leveraging a larger model as a direct evaluator or labeler in such a process relies on its inherent zero-shot evaluation capabilities (Wei et al., 2022a). This use of LLM-generated feedback is conceptually similar to methods like Constitutional AI, where larger models are used to guide smaller ones (Bai et al., 2022). Inspired by these approaches and the prompting techniques of Zheng et al. (2024), our work employs LLMs as labelers for automated process reward annotation, enabling cost-effective synthetic data generation across multiple domains.

## 3. Process Reward Models

Similar to the work of Lightman et al. (2024), we define process rewards to represent the correctness of each step, and whether it is logical and follows from previous steps.

To formally define a PRM, we begin by specifying a CoT

$S = (s_1, s_2, \ldots, s_k)$ as a sequence of $k$ reasoning steps, where $s_i$ is the $i$-th step in the CoT for each $i \in [k]$. A PRM can then be formally characterized as a function that maps each CoT $S$ to an associated $k$-dimensional vector of rewards: $\mathrm{PRM}(S) \in [0,1]^k$. The $i$-th coordinate of the output score vector, denoted as $\mathrm{PRM}(S)_i$, represents the PRM score for the correctness of the reasoning step $s_i$.

### 3.1. Score Aggregation Methods

Using a PRM, we can obtain scores for each reasoning step. To then scalarize the reward score vector of the whole CoT, we consider the following three aggregation methods.

**Min-Aggregation.** We use the minimum PRM step score in a CoT as the aggregated score:
$$\mathrm{Aggr}_{\min}(S) = \min_{i \in [k]} \mathrm{PRM}(S)_i.$$

**Last-Aggregation.** We utilize the PRM score of the last step in a CoT as the aggregated score:
$$\mathrm{Aggr}_{\mathrm{last}}(S) = \mathrm{PRM}(S)_k.$$

**Average-Aggregation.** We employ the average PRM step score of the CoT as the aggregated score:
$$\mathrm{Aggr}_{\mathrm{avg}}(S) = \frac{1}{k} \sum_{i \in [k]} \mathrm{PRM}(S)_i.$$

These aggregated scores are particularly useful for solution reranking and are employed in the test-time inference algorithms described below (Wang et al., 2024b; Sun et al., 2024; Lightman et al., 2024).

### 3.2. Inference-Time Methods

In this section, we introduce three *reranking*-based methods—Majority Voting, Weighted Majority Voting, and Best-of-$N$—along with two *search*-based methods—Beam Search and Monte Carlo Tree Search.

Let $a_S$ denote the final answer in a CoT $S$, which in practice can be extracted using a suitable parser. Further let $\mathcal{S}_N = \{S_1, S_2, \ldots, S_N\}$ denote a set of $N$ CoTs sampled i.i.d. from a generator over a particular question.

**Majority Voting (MV).** MV (Wang et al., 2023) is a robust baseline inference-time method that does not require a PRM. Specifically, we first sample $N$ candidate solutions to a problem from a generator. The final answer is then determined by selecting the solution that appears most frequently among these $N$ candidates:
$$\mathrm{MV}(\mathcal{S}_N) = \underset{a_S : S \in \mathcal{S}_N}{\arg\max} \sum_{i \in [N]} \mathbb{1}_{a_S}(a_{S_i}).$$

**Weighted Majority Voting (WMV).** This method, as used by Uesato et al. (2022), is similar to MV. We sample $N$ candidate solutions. However, we weight the frequencies of CoTs with identical answers by the aggregation scores. The final answer is the one with highest sum of weights:

$$\text{WMV}(\mathcal{S}_N) = \arg\max_{a_S : S \in \mathcal{S}_N} \sum_{i \in [N]} \mathbb{1}_{a_S}(a_{S_i}) \cdot \text{Aggr}(S_i).$$

**Best-of-$N$ (BoN).** This method also samples $N$ candidate solutions. It then reranks them using the aggregation score from a PRM. The answer of the solution with highest score is chosen as final answer:

$$\text{BoN}(\mathcal{S}_N) = \arg\max_{a_S : S \in \mathcal{S}_N} \text{Aggr}(S).$$

**Beam Search.** This method (Snell et al., 2024) is initialized with a fixed number of beams $N$ and width $M$. The process starts by sampling $N$ initial predictions for the first reasoning step. These are ranked via the PRM's step score, retaining the top $\frac{N}{M}$ candidates. For each retained candidate, $M$ proposals for the next step are sampled, yielding $N$ new candidates. This iterates until all beams reach solutions or a maximum iteration limit. The final prediction is selected based on the highest aggregated PRM score across steps. See Algorithm 1 for details.

**Monte Carlo Tree Search (MCTS).** MCTS is a search algorithm used during test-time inference (Hao et al., 2023; Wan et al., 2024) that iteratively builds a search tree to find the CoT with the highest aggregated PRM score. A detailed description is presented in Appendix C and the pseudo-code is provided in Algorithm 2.

# 4. Limitations of Process Reward Models Trained on Math Domain Data

We introduce various math PRMs used for comparison in Section 4.1, present our multi-domain evaluation dataset in Section 4.2, and provide a detailed analysis of the evaluation results in Section 4.3.

## 4.1. Open-Source Math PRMs

For evaluation, we conduct experiments on a diverse set of models. Our analysis includes four open-source math PRMs: Math-PSA (Wang et al., 2024a), Math-Shepherd (Wang et al., 2024b), RLHFLow-Deepseek (Xiong et al., 2024), and Qwen-2.5-Math-PRM (Zheng et al., 2024).

In addition to the open-source models, two math PRMs based on open-source models are specifically trained in this work. They are denoted as *LlamaPRM800K* and *Qwen-PRM800K*. More details are given in Appendix D.1

*Table 1.* Results of two open-source math PRMs on different domains in MMLU-Pro-CoT-Eval (Unlabeled) when using WMV with min-aggregation on 16 CoTs generated per question using Llama-3.1-8B-Instruct. In parenthesis we report absolute difference between WMV and MV (WMV−MV). While WMV using math PRMs exhibits greater improvement in Math and Math-adjacent domains, there is no significant improvement on MV in other domains.

| Category | MV | Math-Shepherd | Qwen-2.5-Math-PRM |
|---|---|---|---|
| All | 57.15 | 57.66 (+0.51) | 58.17 (+1.02) |
| All except math | 56.61 | 57.01 (+0.40) | 57.32 (+0.71) |
| Math | 62.40 | 64.13 (+1.73) | 67.20 (+4.80) |
| Chemistry | 58.67 | 60.13 (+1.46) | 60.67 (+2.00) |
| Physics | 58.53 | 61.87 (+3.34) | 61.47 (+2.94) |
| Biology | 75.38 | 75.38 (+0.00) | 75.69 (+0.31) |
| Psychology | 61.60 | 61.47 (-0.13) | 62.27 (+0.67) |
| Law | 35.93 | 37.24 (+1.31) | 36.28 (+0.35) |
| History | 49.20 | 49.87 (+0.67) | 49.40 (+0.20) |
| Philosophy | 44.83 | 44.70 (-0.13) | 45.17 (+0.34) |

## 4.2. Multi-Domain Evaluation Dataset

For our multi-domain evaluation dataset, we curate questions sampled from the MMLU-Pro dataset (Wang et al., 2024c). MMLU-Pro is designed to benchmark the reasoning abilities of LLMs and consists of college-level multiple choice questions in the following 14 domains: *Math, Physics, Chemistry, Law, Engineering, Other, Economics, Health, Psychology, Business, Biology, Philosophy, Computer Science,* and *History.*

To craft our evaluation dataset, we randomly sample 150 questions from each domain. Due to duplicate questions, we discard 41 questions—23 from Biology, 10 from Health, 5 from Law, and 1 each from Business, Economics, and Philosophy. For each remaining question, we generate 128 candidate solutions using Llama-3.1-8B-Instruct (Dubey et al., 2024) for MV, WMV, and BoN test-time inference algorithms. Prompt details and generation parameters are provided in Appendix A.2. We refer to this multi-domain evaluation dataset as *MMLU-Pro-CoT-Eval (Unlabeled).*

## 4.3. Multi-Domain Performance of Math PRMs

We conduct comprehensive analyses on a diverse set of models. For clarity, we report results for two representative models here, with additional evaluations available in Appendix E.1. The first model, Math-Shepherd (Wang et al., 2024b), is trained on synthetically generated math data labeled via a rollout-based method. The second model, Qwen-2.5-Math-PRM (Zheng et al., 2024), is a best-performing open-source PRM, trained on the high-quality expert labeled PRM800K math dataset (Lightman et al., 2024).

The PRMs are applied using WMV with min-aggregation. While math PRMs show significant improvements in mathematical reasoning domains, their effectiveness in broader, non-mathematical areas remains limited. Notably, in the Math category, Qwen-2.5-Math-PRM and Math-Shepherd achieve relative gains of +4.80 and +1.73, respectively, outperforming the MV baseline. Similar improvements are observed in Math-adjacent disciplines: Chemistry (+2.00 for Qwen-2.5-Math-PRM) and Physics (+3.34 for Math-Shepherd), underscoring their utility in tasks requiring mathematical reasoning.

> **Finding 1:** *Math PRMs struggle to generalize to broader domains.*

However, the benefits diminish sharply in non-mathematical areas. For example, in Philosophy and History, we see gains of only +0.34 and +0.20% respectively for the most performant PRM Qwen-2.5-Math-PRM.

The "All except math" aggregate further underscores this disparity, with PRMs achieving a maximum gain of +0.71 (Qwen-2.5-Math-PRM) compared with the majority voting baseline.

These results highlight a critical limitation: math PRMs trained exclusively on mathematical data lack the versatility to generalize beyond mathematical reasoning tasks. While they excel in contexts aligned with their training—quantitative reasoning—their capacity to evaluate reasoning quality in broader domains remains insufficient.

## 5. Automatic Generation of Multi-Domain Reasoning Data with Labels

In order to obtain step-wise reasoning data for non-Math domains, we devise a pipeline, as outlined in Figure 2, to generate synthetic reasoning CoTs from existing question-answering data. These CoTs are then given step-wise labels based on reasoning correctness. We detail the synthetic data generation process in Sections 5.1 and 5.2, including methods to create and annotate reasoning steps. We also provide additional analysis on the quality of the generation pipeline in Section 5.3.

### 5.1. Chain-of-Thought Generation

For the generation of CoTs, we prompt Llama-3.1-8B-Instruct to produce step-by-step reasoning for each input question. For training, we source questions from the MMLU-Pro dataset (Wang et al., 2024c), by randomly sampling up to 500 questions per domain, ensuring that it is disjoint to the subset used for evaluation. We then generate 16 CoTs for each sampled question. Post-generation, we filter out CoTs exceeding the 2,048-token limit or containing unparsable answers.

### 5.2. Auto-Labeling

To annotate our synthetic CoT data, we adopt an approach inspired by the critic models in the work of Zheng et al. (2024). Specifically, we utilize Llama-3.1-70B-Instruct as a strong LLM to evaluate each CoT using step-by-step reasoning, locating the earliest erroneous step, if any. To enhance accuracy and consistency, we identified two key additional components.

First, we incorporate explicit step evaluation definitions, inspired by Lightman et al. (2024), into the system prompt. Steps are categorized as **GOOD**, **OK**, or **BAD**: **BAD** for incorrect, unverifiable, or irrelevant steps; **GOOD** for correct, verifiable, and well-aligned steps; **OK** for intermediate cases. Second, we also provide the ground-truth reference answer for the question whose CoT is being graded in the prompt. The full prompt is detailed in Appendix A.2.

To convert the auto-labeling output to stepwise labels, we apply the following rule: if no steps are detected as incorrect, all steps in the CoT are labeled as $1$. If a step is detected as incorrect, all preceding steps are labeled as $1$, the incorrect step is labeled as $-1$, and all subsequent steps are discarded.

In total, we sample 5,750 questions from MMLU-Pro. Among the 84,098 generated CoTs that passed filtering, 36,935 were labeled as having no incorrect steps and 47,163 were labeled as having at least one (see Table 2). The CoTs generation and labeling were done using AWS Bedrock batch inference at a total cost under $100 (USD). This dataset, denoted as *MMLU-Pro-CoT-Train (Labeled)*, is the first open-source multi-domain reasoning dataset with step-wise labels.

To assess the quality of our auto-labeled data, we conduct a manual evaluation on a random sample of 64 questions from the dataset. For each question, we randomly select one CoT classified as entirely correct and two CoTs flagged as containing an incorrect step. We then manually validate whether the auto-labeled judgments align with our own assessments.

For the CoTs labeled as correct by the auto-labeler, we observed an agreement rate of 80% (95% CI: 0.69–0.89) with our manual evaluations. For CoTs labeled as incorrect, the agreement rate was 71% (95% CI: 0.63–0.79).

Based on these results, we estimate that approximately 75% of the CoTs in the entire dataset are correctly labeled. This level of accuracy is comparable to that of manually-labeled CoT datasets, such as PRM800K (Lightman et al., 2024), which is estimated to achieve around 80% accuracy.[1]

---

[1] See the following Github issue for a discussion: `https://github.com/openai/prm800k/issues/12`.

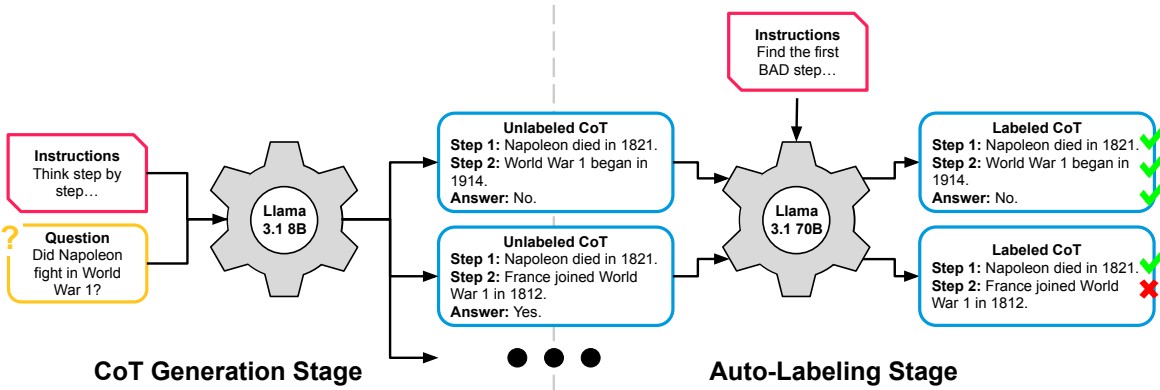

*Figure 2.* A diagram of the synthetic data generation pipeline. In the CoT Generation Stage, each question is used to generate 16 CoT solutions. Then, in the Auto-Labeling Stage, each CoT is evaluated to create step-wise labels. If a CoT step is labeled as **BAD**, all subsequent steps will be discarded.

### 5.3. Auto-Labeling Prompt Analysis

To further understand the factors influencing auto-labeling performance, we conduct an evaluation of the auto-labeling using a simplified prompt. Specifically, we remove the system prompt defining the types of reasoning steps and exclude the reference ground-truth answer from the prompt. When re-evaluating the auto-labeling quality on a random subset of 30 questions from the original sampled 64, we observe a drastic drop in performance, with the agreement rate for CoTs labeled as correct by the original auto-labeler decreasing by over 70%, to 7%, while the agreement rate for CoTs labeled as incorrect decreased from 71% to 62%. Additionally, in Appendix B.4, we provide an end-to-end experiment that evaluates a PRM trained on data labeled using this ablated prompt. The findings mirror our manual evaluation—the PRM trained with the ablated prompt used via BoN achieves less than half of the performance uplift over MV that the full model attains.

These results highlight the importance of providing both a well-defined prompt with step label definitions and access to the ground-truth answer in achieving high auto-labeling accuracy. The ground-truth answer provides essential context on CoT final correctness and enhances the model's ability to evaluate reasoning steps effectively.

### 5.4. Synthetic Data Augmentations

To further augment the dataset, we experiment with two synthetic augmentation methods. The first, which we term *counterfactual augmentation*, involves generating additional examples of incorrect reasoning by prompting an LLM to modify steps in correct CoTs, thereby introducing targeted errors. The second method is a straightforward rewriting approach, where an LLM rewrites a step to preserve its original meaning but alter its style.

However, incorporating augmentations from either method during PRM training did not lead to significant performance improvements. As a result, we defer the specific details and experiments on counterfactual augmentation and rewrite augmentation to Appendices B.1 and B.2, respectively.

## 6. Multi-Domain Process Reward Model

We present the implementation and evaluation of VersaPRM, structured as follows. First, Section 6.1 covers the various training configurations used. We then evaluate VersaPRM via BoN and WMV in Section 6.2, showing improved domain generalization compared to math PRMs. In Section 6.3, we additionally discuss results using Beam Search and MCTS. Lastly, we examine VersaPRM's ability to scale test-time compute for larger models such as Deepseek-R1 (Guo et al., 2025) in Section 6.4.

### 6.1. Training of Our Multi-Domain PRM

To train VersaPRM, we employ a classification head atop an LLM, optimizing with a cross-entropy loss applied to a special classification token appended at the end of each CoT step in MMLU-Pro-CoT-Train (Labeled). Detailed specifics and hyperparameters are provided in Appendix D.2.

We explore several training configurations, including: 1) LoRA (Hu et al., 2022) vs. full fine-tuning for efficient training, 2) a base LLM vs. a math PRM for initializing the PRM, and 3) a Qwen-based PRM vs. a Llama-based PRM for training. Comprehensive experimental results for these studies are presented in the next section. Based on those findings, our final, our final multi-domain PRM, named VersaPRM, is initialized from our LlamaPRM800K—see Appendix D for its details—fine-tuned using LoRA on our multi-domain training dataset.

## 6.2. Math PRM vs. VersaPRM on Reranking Based Inference-Time Methods

We first report results of the reranking methods WMV and BoN on MMLU-Pro-CoT-Eval (Unlabeled). For both methods, we adopt Min-aggregation, as it outperforms Average and Last in aggregating PRM step scores; see Appendix E.3 for comparison. We also include MV as a baseline.

**Comparison with Math Open-Source PRMs.** We evaluate our multi-domain PRM, VersaPRM, against open-source math PRMs by partitioning MMLU-Pro-CoT-Eval (Unlabeled) into three groups: 1) *Math*, 2) *Math-adjacent*, i.e., Chemistry, Computer Science, Engineering, Physics, and 3) *non-Math-adjacent* domains. As shown in Figure 3, our model consistently outperforms baselines in both WMV and BoN across all domain groups.

> **Finding 2:** *Fine-tuning with synthetic multi-domain data enhances the PRM's performance in non-math domains.*

For WMV, we can see the relative performance difference increase with domain distance from core mathematics. While performance of math PRMs converges to the majority voting baseline in non-mathematical domains, VersaPRM maintains robust performance in these other domains.

In BoN the superiority of VersaPRM is even more pronounced. Unlike math PRMs, which fail to surpass the baseline of MV in Math-adjacent and non-Math-adjacent domains, our model consistently surpasses it across all domain groups.

See Appendix E.5 for more fine-grained details where we plot WMV and BoN for every domain of MMLU-Pro-CoT-Eval (Unlabeled). The results are consistent with Figure 3, and VersaPRM outperforms math PRMs in all domains.

**Ablation Experiments Using VersaPRM Trained on Math Only Subset vs. Random Subset.** We further conduct an ablation study to evaluate the impact of training data diversity on the performance of our LlamaPRM800K Math PRM. Specifically, we train one PRM using only the math subset of our multi-domain training data and another using a random subset of the *same* size. We refer to these two models as VersaPRM (Math subset) and VersaPRM (random subset), respectively. This experiment tests that the improved performance of our multi-domain PRM is due to the domain-diversity of the CoT data and not merely from learning the in-distribution question and CoT formats of MMLU-Pro questions. If the latter is the case, both PRMs should perform similarly, given that they are exposed to the same amount of questions and CoT examples with the in-distribution format.

> **Finding 3:** *Domain diversity of CoTs in a training dataset plays an integral role in generalization of PRMs to multiple domains.*

As shown in Figure 4, VersaPRM (Math subset) obtains a modest lift over LlamaPRM800K—evidence that learning the in-domain question format partially helps. Far more striking is that VersaPRM (random subset) obtains markedly higher WMV accuracy across both math and non-math tasks These findings suggest two key insights. First, our PRM is not simply learning the question format but is acquiring knowledge on how to label reasoning across diverse domains. This is why training on diverse data enables better overall performance than training on same sized data in only one domain. Second, VersaPRM (random subset) also demonstrates slightly better performance in the math domain, indicating that training on a diverse dataset may facilitate positive transfer from other domains to math.

**Ablation Experiments using a Hold-out Domain.** In order to verify the domain-general reasoning capabilities of VersaPRM, we conduct additional ablation experiments using a hold-out domain approach. Specifically, we excluded one domain category (law or biology) from the VersaPRM training set and evaluate the model's performance on the held-out domain. As illustrated in Figure 5, the performance of VersaPRM trained with one domain held out remains comparable to the fully trained model across both evaluated domains. Additionally, they both surpass the performance of the math PRMs. These results underscore that the generalization ability of VersaPRM is not merely due to broader coverage in the training data, but rather represents genuine domain-general reasoning capabilities. Additional results on other domains are presented in Appendix E.7.

**Experiments Using Other Training Configurations.** While our final version of VersaPRM is trained from LlamaPRM800K on our synthetic data using LoRA, we also test the following training configurations on our multi-domain dataset:

- **VersaPRM (Llama Base)**: We initialize training from Llama-3.1-8B-Instruct, and use LoRA fine-tuning with our multi-domain dataset.

- **VersaPRM (Qwen)**: We initialize training from Qwen-PRM800K PRM, and utilize LoRA fine-tuning with our multi-domain dataset.

- **VersaPRM (full-tuned)**: We initialize training from LlamaPRM800K PRM, and do *full* fine-tuning with our multi-domain dataset.

The results are presented in Figure 6. Comparing VersaPRM (Qwen) and VersaPRM (Llama), we observe that

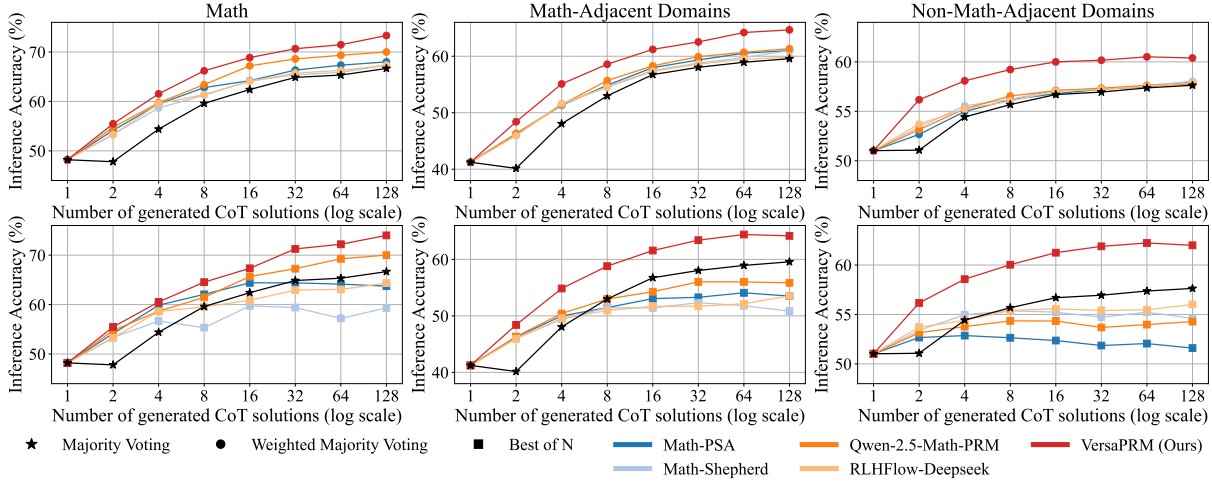

*Figure 3.* Comparison of WMV (top) and BoN (bottom) using VersaPRM against open-source math PRMs on MMLU-Pro-CoT-Eval (Unlabeled). We use min-aggregation and the CoTs are generated using Llama-3.1-8B-Instruct. VersaPRM has consistently better performance than math PRMs, and the differences become larger in domains not adjacent to Math.

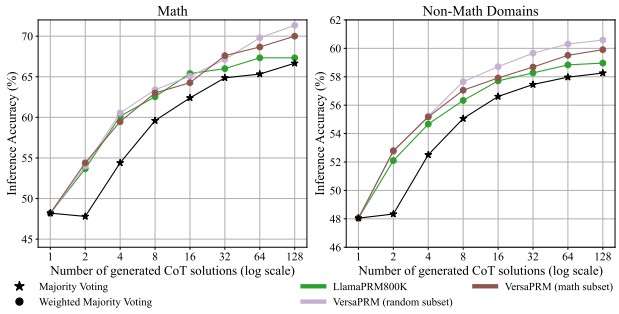

*Figure 4.* Comparison of WMV using LlamaPRM800K, VersaPRM (Math subset) and VersaPRM (random subset). VersaPRM (random subset) achieves better performance than VersaPRM (Math subset) in Math and non-Math.

*Figure 5.* Hold-out domain ablation experiment results. WMV performance comparison of VersaPRM trained with each of the indicated domains held out versus the fully trained VersaPRM.

the QwenPRM800K VersaPRM performs worse. This highlights the importance of base model choices. Although Qwen-2.5-Math-7B, the base model for QwenPRM800K, is specialized in mathematical reasoning, its limitations in general-domain knowledge hinder its ability to fully leverage multi-domain training data.

> **Finding 4:** *Exposure to mathematical data beforehand can enhance a PRMs' ability to effectively leverage multi-domain CoT fine-tuning.*

Next, comparing VersaPRM (Llama Base) with VersaPRM, we find that the latter achieves superior performance in Math while maintaining comparable performance in non-Math domains. This suggests that prior exposure to mathematical data enhances the model's ability to benefit from further

domain-specific training.

We note that VersaPRM (full-tuned) has worse performance than VersaPRM. This may be due to suboptimal hyperparameters leading to overfitting during full fine-tuning.

### 6.3. Math PRM vs. Multi-Domain PRM on Search Based Inference-Time Methods

We evaluate the performance of math PRMs (using LlamaPRM800K) and VersaPRM with beam search and MCTS on MMLU-Pro-CoT-Eval (Unlabeled). The results over questions in all domains, presented in Figure 7, show that MCTS outperforms beam search and that they both do better than the MV baseline. Regardless of the search algorithm, consistent with our WMN and BoN results, VersaPRM gives boosted performance over the math PRM. It is also interesting to note that the performance of these search

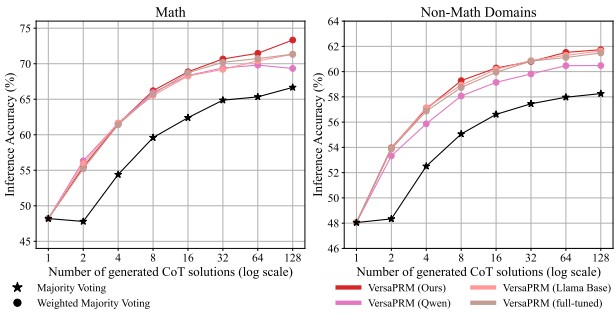

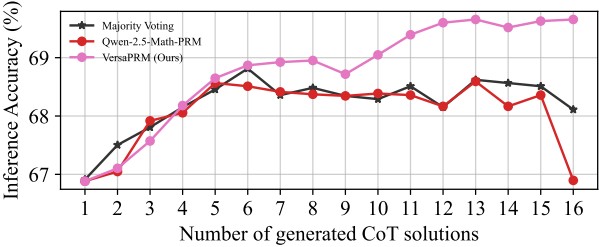

(a) Law Domain

*Figure 6.* Comparison of MVW using VersaPRM against other multi-domain PRMs trained using different configurations. VersaPRM has better WMV performance than all other models in both Math and non-Math domains.

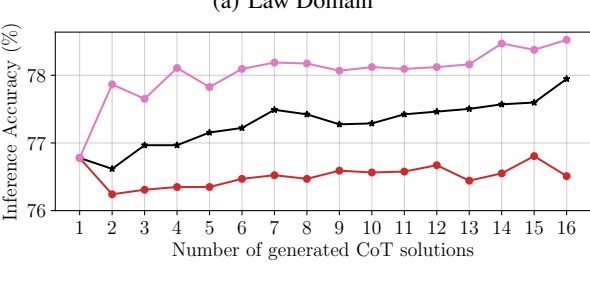

(b) Philosophy Domain

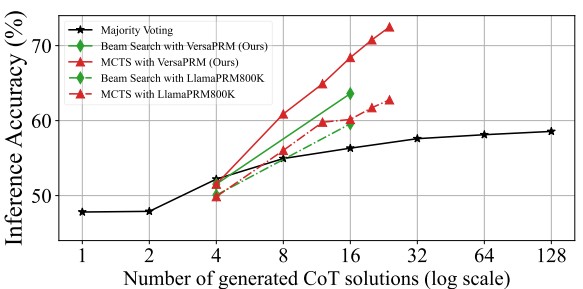

*Figure 7.* Comparison of VersaPRM and LlamaPRM800K with beam search and MCTS. The x-axis compares MCTS and Beam Search in terms of computational cost for an equivalent number of generated CoT solutions. Overall in the diverse domains from MMLU-Pro-CoT-Eval (Unlabeled), VersaPRM achieves better performance.

*Figure 8.* Comparison of WMV using VersaPRM against Qwen-2.5-Math-PRM and Majority Voting on DeepSeek-R1 generated CoTs for Law and Philosophy subsets. VersaPRM consistently outperforms the math PRM and MV baseline in both subsets.

algorithms saturate much slower than WMN and BoN. Details by category results are presented in Appendix E.6.

### 6.4. Does PRM with Test-Time Compute help Reasoning Models?

Having shown that VersaPRM effectively leverages inference-time compute to enhance LLM performance, we next examine whether this benefit extends to strong reasoning models such as DeepSeek-R1 (Guo et al., 2025). Given DeepSeek-R1's extensive reasoning-focused training, one might hypothesize that reranking methods (e.g., WMV and BoN) offer minimal additional improvements.

To test this, we evaluate VersaPRM using WMV on DeepSeek-R1 for the Law and Philosophy subsets, sampling 16 CoT responses per question. As illustrated in Figure 8, VersaPRM achieves a modest yet clear improvement over both the math PRM and MV baselines. These preliminary results counter the hypothesis and indicate that even highly capable reasoning models *can* benefit from PRM-enhanced

inference.

## 7. Discussion and Future Directions

We proposed VersaPRM trained using synthetic reasoning data to address the limitations of existing math PRMs. By leveraging a cost-efficient synthetic data generation pipeline, we enabled production of high-quality step-wise reasoning data and demonstrate that PRMs can effectively scale reasoning ability at inference time in diverse domains.

**Future Work.** Several directions remain for advancing multi-domain PRMs. Can VersaPRM be effectively used as a reward model for reinforcement learning (RL) training? Can VersaPRM also improve RL training in domains beyond mathematics? Could more sophisticated counterfactual augmentation enhance PRM effectiveness? From an alignment perspective, given that both the generator and labeler LLMs are Llama models, could VersaPRM be biased towards performing better with Llama-based generator models? Furthermore, evaluating PRMs on more challenging, open-ended problems, and on distinct task categories such as puzzles or games, would better assess their generalization capabilities. Finally, a more thorough investigation of PRMs with larger models (e.g., GPT-4 (Achiam et al., 2023) and DeepSeek-R1) could clarify their scalability and potential role in state-of-the-art reasoning systems.

## Acknowledgements

Kangwook Lee is supported by NSF Award DMS-2023239, NSF CAREER Award CCF-2339978, an Amazon Research Award, and a grant from FuriosaAI. In addition, Thomas Zeng acknowledges support from NSF Award DMS-2023239 and Daewon Chae is supported by the Hyundai Motor Chung Mong-Koo Foundation.

## Impact Statement

Given the potential for LLMs to be used in unethical ways, such as spreading misinformation or manipulating public opinion, VersaPRM could inadvertently contribute to such misuse. To mitigate these risks, it is essential to implement robust safeguards in training and inference.

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

# A. More Details on Synthetic Data Generation Pipeline

## A.1. Dataset Composition

The total composition of MMLU-Pro-CoT-Train (Labeled) is as follows.

*Table 2.* Composition of *MMLU-Pro-CoT-Train (Labeled)*

|  | **Total** | **Fully Correct** | **Incorrect** |
| --- | --- | --- | --- |
| Number of CoTs | 84,098 | 36,935 | 47,163 |
| Number of Steps | 487,380 | 440,217 | 47,163 |

## A.2. Data Generation Pipeline Prompts

To generate chain-of-thought (CoT) reasoning for MMLU-Pro questions, we utilize the prompt shown in Figure 12. To ensure the generated CoT adhere to the proper format—where steps are separated by two newline characters and the final step follows the structure "the answer is (X)"—we include five few-shot examples. These examples are derived from the CoTs provided in the validation split of MMLU-Pro, with additional processing to ensure each step is delimited. The code for generating the complete prompt will be open-sourced alongside the rest of our code and data.

During generation, we use a temperature of $0.8$ and set the maximum generation length to 2,048 tokens. During auto-labeling, we use a temperature of 0, and the maximum generation length remains at 2,048 tokens.

The following is a multiple choice question and its ground truth answer. You are also given a students solution (split into step, enclosed with tags and indexed from 0):

[Multiple Choice Question]
{question}

[Ground Truth Answer]
{answer}

[Student Solution]
{<step_0>
Student solution step 0
</step_0>

<step_1>
Student solution step 0
</step_1>

...}

*Figure 9.* User prompt template for auto-labeling.

You are an experienced evaluator specializing in assessing the quality of reasoning steps in problem-solving. Your task is to find the first BAD step in a student's solution to a multiple choice question.

You will judge steps as GOOD, OK or BAD based on the following criteria:
1. GOOD Step
A step is classified as GOOD if it meets all of these criteria:
- Correct: Everything stated is accurate and aligns with known principles or the given problem.
- Verifiable: The step can be verified using common knowledge, simple calculations, or a quick reference (e.g., recalling a basic theorem). If verifying requires extensive effort (e.g., detailed calculations or obscure references), mark it BAD instead.
- Appropriate: The step fits logically within the context of the preceding steps. If a prior mistake exists, a GOOD step can correct it.
- Insightful: The step demonstrates reasonable problem-solving direction. Even if ultimately progress in the wrong direction, it is acceptable as long as it represents a logical approach.

2. OK Step
A step is classified as OK if it is:
- Correct and Verifiable: Contains no errors and can be verified.
- Unnecessary or Redundant: Adds little value, such as restating prior information or providing basic encouragement (e.g., "Good job!").
- Partially Progressing: Makes some progress toward the solution but lacks decisive or significant advancement.

3. BAD Step
A step is classified as BAD if it:
- Is Incorrect: Contains factual errors, misapplies concepts, derives an incorrect result, or contradicts the ground truth answer.
- Is Hard to Verify: Requires significant effort to confirm due to poor explanation.
- Is Off-Topic: Includes irrelevant or nonsensical information.
- Derails: Leads to dead ends, circular reasoning, or unreasonable approaches.

#### Task Description
You will be provided with:
1. A Question
2. A Ground Truth Answer
3. A Reference explanation of the answer
4. A Student's Step-by-Step Solution, where each step is enclosed with tags and indexed from 0

You may use the ground truth answer and reference explanation in classifying the type of each step. A student's final answer is considered correct if it matches the ground truth answer or only differs due to differences in how the answer is rounded. Once you identify a BAD step, return the index of the earliest BAD step. Otherwise, return the index of -1 (which denotes all steps are GOOD or OK). Please put your final answer (i.e., the index) in \\boxed.

*Figure 10.* System prompt for auto-labeling.

# B. Additional Training Experiments and Ablations

In this section we detail additional training and ablation experiments for VersaPRM. The final evaluation results of these methods are presented in Appendix B.6.

### B.1. Counterfactual Augmentation

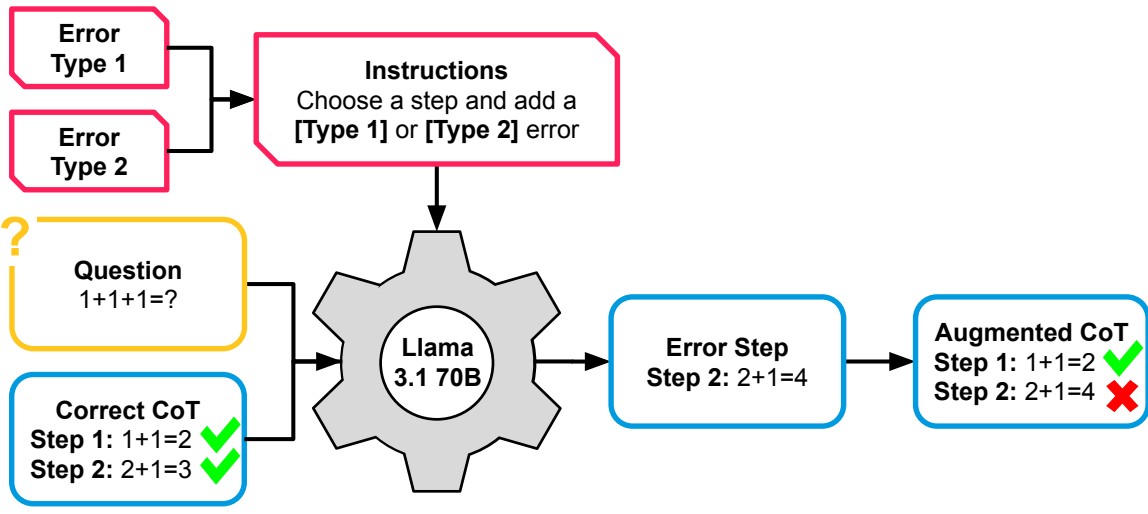

*Figure 11.* Diagram of the counterfactual augmentation pipeline

After generating and labeling our synthetic reasoning CoTs (as described in Section 5), we attempted to create additional incorrect steps by augmenting the correct reasoning steps. Our pipeline is depicted in Figure 11. We provide the full CoT to Llama-3.1-70B-Instruct, instructing it to select and rewrite a step where it would be appropriate to introduce an error. Additionally, we define a list of possible fine-grained error types. To encourage the generation of a variety of different error types, we only include a random selection of two of these error types in each system prompt, forcing the LLM to choose one. The error types are:

- Conflicting Steps: The reasoning step includes statements that contradict previous steps.

- Non-sequitur: The reasoning step introduces information that is irrelevant to the question.

- Factual: The reasoning step contains incorrect statements of factual information.

- False Assumption: The reasoning step makes an incorrect assumption about the question.

- Contextual: The reasoning step misinterprets information given within the question/context.

For the prompt format used in counterfactual augmentation, see Figure 13. In total, we generated 73,829 augmented incorrect steps.

> The following are multiple choice questions (with answers). Think step by step and then finish your answer with "the answer is (X)" where X is the correct letter choice.

*Figure 12.* Prompt to generate CoTs for MMLU Pro.

You are a highly knowledgeable philosopher with expertise across many domains, tasked with analyzing reasoning processes. Your goal is to identify how a reasoning process could naturally deviate toward an incorrect conclusion through the introduction of subtle errors.

Here are a list of potential error types, all of which are equally valid:
[ERROR TYPE 1]: [ERROR TYPE 1 DEFINITION]
[ERROR TYPE 2]: [ERROR TYPE 2 DEFINITION]

Instructions:
You will be provided with:
1. A question.
2. A complete chain of reasoning steps, where each step is numbered (e.g., Step X).

Your task is to: 1. Identify the major factual information, reasoning, and conclusions within the reasoning steps.
3. Explain how to generate an incorrect step to replace one of the existing steps. This should include:
- Identifying a step where the reasoning could naturally deviate.
- Speculating what type of error would be most appropriate to introduce at the chosen step.
4. Introduce an incorrect next step that aligns stylistically with the previous steps. This incorrect step should:
- Reflect a deviation in reasoning that significantly harms the correctness.
- Appear natural and believable in the context of the reasoning process.
5. Clearly explain how the incorrect step is an error, highlighting the specific logical or conceptual flaw.

Output Format:

STEP_SUMMARY:
[Summarize the reasoning within the steps in 1-2 sentences, identifying major information, logical steps, and conclusions.]

INCORRECT_STEP_GEN:
[Explain how the reasoning at a specific step could deviate naturally into being incorrect. Clearly describe the type of error that could be introduced at this step.]

ERROR_TYPE:
[The name of the type of error chosen to be introduced.]

STEP_NUM:
[The number of the step that was identified as a place where the reasoning could naturally deviate. Only include the number here.]

INCORRECT_STEP:
[Write the incorrect step in the same tone and style as the other steps. Wrap the incorrect step inside curly braces (e.g. {incorrect step}).]

ERROR_EXPLANATION:
[Explain how the incorrect step fits the definition of the selected error type, identifying the specific flaw.]

*Figure 13.* System prompt for counterfactual augmentation.

## B.2. Rewrite Augmentation

To further enrich the training dataset and enhance model robustness, we implemented rewrite augmentation. We randomly sampled one intermediate reasoning step (excluding the final answer step) from each CoT in the labeled MMLU-Pro-CoT-Train dataset and tasked Llama-3.1-70B-Instruct with rewriting the step to preserve its logical content while varying its wording.

The prompt for rewrite augmentation explicitly required substantial rephrasing without introducing or omitting critical reasoning content (see Figure 14).

---

You will be given a reasoning step from a larger chain of thought. Your task is to rewrite this step using different phrasing, while keeping the underlying reasoning and deduction the same.
The rewritten step must:
- Preserve the same logical content and conclusion
- Match the tone and level of formality of the original
- Use different wording and phrasing, not just minor edits or clause reordering
Do not introduce new information or omit key reasoning.
Format your output as:

<rewritten_step>your rewritten version here</rewritten_step>

### Original Step:

`{original reasoning step}`

---

*Figure 14.* Prompt to generate rewrite augmentations.

## B.3. Self-Filtering

Motivated by prior works on self-training (Amini et al., 2025) and self-filtering (Shen & Sanghavi, 2019), we applied a self-filtering procedure to the labeled MMLU-Pro-CoT-Train dataset. Specifically, we utilized VersaPRM to assign scores to each step within the dataset and filtered out entire CoTs containing any step whose predicted score deviated by more than 0.4 from its autolabeled score. This self-filtering process resulted in removing approximately 37% of CoTs from the original dataset. We then train a PRM on the resulting filtered dataset.

## B.4. Ablation Study on Inclusion of Ground Truth in Auto-Labeling Prompt

As an additional ablation, we do an end-to-end experiment evaluating the impact of omitting the ground truth answer from the auto-labeling prompts. We performed an end-to-end experiment by auto-labeling the entire MMLU-Pro-CoT-Train dataset using a modified prompt that explicitly excluded the correct answer. Subsequently, we trained a variant of VersaPRM on this modified dataset.

The detailed ablated prompt used for autolabeling (excluding the ground truth answer) is provided in Figure 15.

## B.5. Llama-as-a-Judge Baseline

Finally, we considered an additional baseline leveraging Llama-3.1-70B-Instruct directly as a process reward model (PRM). Here, we used Llama-3.1-70B-Instruct with the ablated auto-labeling prompt (see Figure 15) to identify the earliest incorrect step in each CoT. Scores of 1 were assigned to steps preceding the identified incorrect step, while scores of 0 were assigned to the incorrect step and all subsequent steps. Importantly, this approach did not utilize the ground truth answer at inference, reflecting a realistic scenario.

## B.6. Experimental Results

We evaluate all of the methods presented in this section against VersaPRM using both BoN (Figure 16) and WMV (Figure 17). For each metric we apply min-aggregation across all categories of MMLU-Pro-CoT-Eval (Unlabeled).

We find that the training enhancements—counterfactual augmentation, rewrite augmentation, and self-filtering—provide no

significant benefit over VersaPRM. Under BoN, VersaPRM attains the best aggregate performance across categories, while under WMV the enhancements offer only marginal gains, and only at large values of $N$.

Aggregated over all categories, VersaPRM also outperforms (i) directly using Llama 3.1-70B as a judge and (ii) the end-to-end variant of VersaPRM trained on data from the ablated prompt without ground-truth labels, for both BoN and WMV at every value of $N$. These results underscore that including the ground-truth answer in the labeling prompt is essential for the autolabeler to assign labels accurately.

You are an experienced evaluator specializing in assessing the quality of reasoning steps in problem-solving. Your task is to find the first BAD step in a student's solution to a multiple choice question.

You will judge steps as GOOD, OK or BAD based on the following criteria:
1. GOOD Step
A step is classified as GOOD if it meets all of these criteria:
- Correct: Everything stated is accurate and aligns with known principles or the given problem.
- Verifiable: The step can be verified using common knowledge, simple calculations, or a quick reference (e.g., recalling a basic theorem). If verifying requires extensive effort (e.g., detailed calculations or obscure references), mark it BAD instead.
- Appropriate: The step fits logically within the context of the preceding steps. If a prior mistake exists, a GOOD step can correct it.
- Insightful: The step demonstrates reasonable problem-solving direction. Even if ultimately progress in the wrong direction, it is acceptable as long as it represents a logical approach.

2. OK Step
A step is classified as OK if it is:
- Correct and Verifiable: Contains no errors and can be verified.
- Unnecessary or Redundant: Adds little value, such as restating prior information or providing basic encouragement (e.g., "Good job!").
- Partially Progressing: Makes some progress toward the solution but lacks decisive or significant advancement.

3. BAD Step
A step is classified as BAD if it:
- Is Incorrect: Contains factual errors, misapplies concepts, derives an incorrect result, or contradicts the ground truth answer.
- Is Hard to Verify: Requires significant effort to confirm due to poor explanation.
- Is Off-Topic: Includes irrelevant or nonsensical information.
- Derails: Leads to dead ends, circular reasoning, or unreasonable approaches.

#### Task Description
You will be provided with:
1. A Question
2. A Ground Truth Answer

Once you identify a BAD step, return the index of the earliest BAD step. Otherwise, return the index of -1 (which denotes all steps are GOOD or OK). Please put your final answer (i.e., the index) in \\boxed.

*Figure 15.* Ablated system prompt used for auto-labeling where the ground truth answer is not provided.

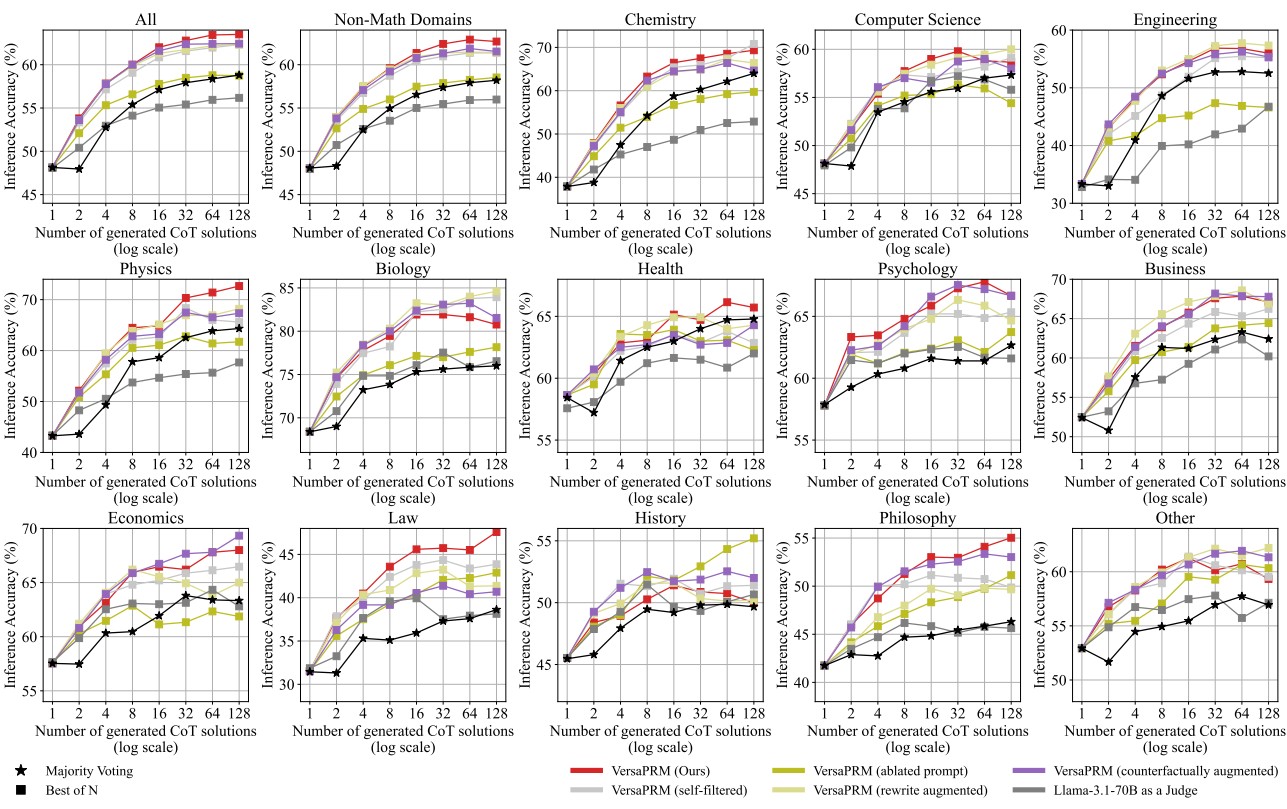

*Figure 16.* Comparison of BoN using VersaPRM against the methods proposed in Appendix B over all categories of MMLU-Pro-CoT-Eval (Unlabeled). We use min-aggregation and the CoTs are generated using Llama-3.1-8B-Instruct.

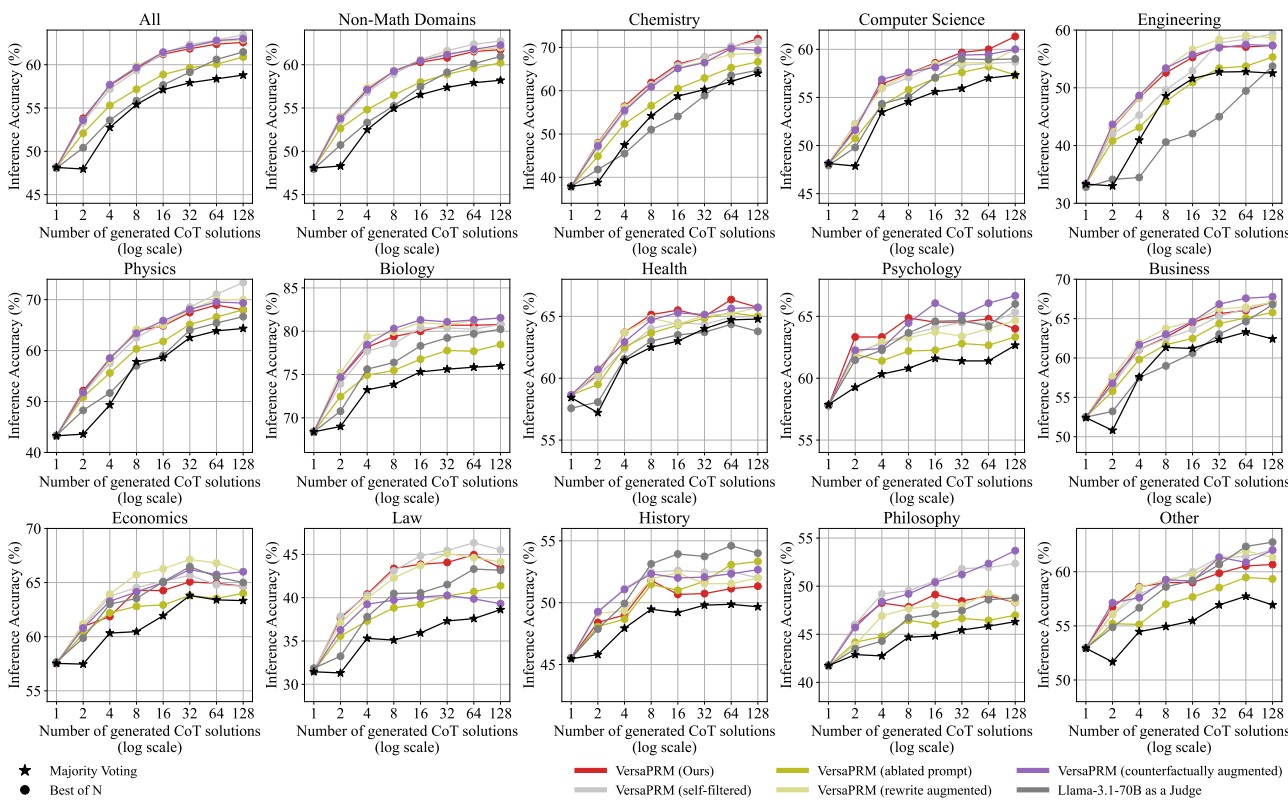

*Figure 17.* Comparison of WMV using VersaPRM against the methods proposed in Appendix B over all categories of MMLU-Pro-CoT-Eval (Unlabeled). We use min-aggregation and the CoTs are generated using Llama-3.1-8B-Instruct.

# C. Additional Search Algorithm Details

---

**Algorithm 1** Beam Search with Process Reward Model

---

**Require:** Large Language Model LLM$(\cdot)$, Process Reward Model PRM$(\cdot)$, Prompt $s_0$, Number of Beams $N$, Beam width $M$, Maximum step length $L$

1: $\mathcal{B} \leftarrow [s_0]$
2: $\mathcal{Q} \leftarrow [0]$
3: **for** $i = 1$ to $L$ **do**
4:      $\mathcal{B} \leftarrow \text{Expand}(\mathcal{B}, \frac{N}{\text{len}(\mathcal{B})})$
5:      $\mathcal{B} \leftarrow \text{LLM.step}(\mathcal{B})$
6:      $\mathcal{Q} \leftarrow \text{Aggr}(\mathcal{B})$
7:      $\texttt{best\_idxs} \leftarrow$ Indexes of the highest $\frac{N}{M}$ scores in $\mathcal{Q}$
8:      $\mathcal{B} \leftarrow \mathcal{B}[\texttt{best\_idxs}]$
9:      $\mathcal{Q} \leftarrow \mathcal{Q}[\texttt{best\_idxs}]$
10:     **if** All sequences in $\mathcal{B}$ contain a terminal leaf node **then**
11:        **break**
12:     **end if**
13: **end for**
14: Return the sequence with the highest score from $\mathcal{B}$

---

Algorithm 1 is a greedy search algorithm that uses a PRM select the best CoT during search. More details are given in Section 3.2.

---

**Algorithm 2** Monte Carlo Tree Search with Process Reward Model

---

**Require:** Large Language Model LLM($\cdot$), Process Reward Model PRM($\cdot$), Prompt $s_0$, Maximum step length $L$, Number of roll-outs $K$, Number of generated child nodes $d$, Exploration weight $w$

1: Initialize the value function $Q : \mathcal{S} \mapsto \mathbb{R}$ and visit counter $N : \mathcal{S} \mapsto \mathbb{N}$
2: **for** $n \leftarrow 0, \ldots, K-1$ **do**
3:     // *Selection*
4:     $t \leftarrow 0$
5:     **while** $s_t$ is not a leaf node **do**
6:         $N(s_t) \leftarrow N(s_t) + 1$
7:         $s_{t+1} \leftarrow \arg\max_{\text{children}(s_t)} \left[ Q(\text{child}(s_t)) + w\sqrt{\frac{\ln N(s_t)}{N(\text{child}(s_t))}} \right]$
8:         $t \leftarrow t + 1$
9:     **end while**
10:     // *Expansion & Simulation* (equivalent to the beam search with $N = M = d$)
11:     $\mathcal{B} \leftarrow [s_t]$
12:     **while** $s_t$ is not a terminal leaf node $\wedge\ t \leq L$ **do**
13:         $N(s_t) \leftarrow N(s_t) + 1$
14:         $\mathcal{B} \leftarrow \text{Expand}(\mathcal{B}, d)$
15:         $\mathcal{B} \leftarrow \text{LLM.step}(\mathcal{B})$
16:         **for** $s \in \mathcal{B}$ **do**
17:             $Q(s) \leftarrow \text{Aggr}(s)$
18:         **end for**
19:         $s_{t+1} \leftarrow \arg\max_{s \in \mathcal{B}} Q(s)$
20:         $t \leftarrow t + 1$
21:         $\mathcal{B} \leftarrow [s_t]$
22:     **end while**
23:     // *Back Propagation*
24:     **for** $t' \leftarrow t, \ldots, 0$ **do**
25:         $Q(s_{t'}) \leftarrow \max(Q(s_{t'}), Q(s_t))$
26:     **end for**
27: **end for**
28: Return the sequence with the highest score among the terminal nodes

---

Algorithm 2 is a tree-based search algorithm that iteratively expands a search tree to find the CoT with the highest PRM score. MCTS iteratively builds a search tree through the following steps:

1. **Selection**: Starting from the root node, the algorithm traverses the tree by selecting child nodes according to a selection policy.

2. **Expansion and Simulation**: Upon reaching a non-terminal leaf node, the tree is expanded iteratively by generating a fixed number of child nodes and then greedily selecting the child node with the highest value (which for us is determined by the PRM). This process continues until a terminal node is reached.

3. **Backpropagation**: The results from the simulation are propagated back through the tree, updating value estimates and visit counts for each node along the path.

These steps are repeated for a fixed number of iterations or until a computational or time limit is reached. To determine the final prediction, we choose the terminal node with the highest value.

# D. Additional PRM Training Details

## D.1. Open-Source Math PRM Training Details

The open-source PRMs evaluated in this work utilize CoT training data derived from two mathematical datasets: MATH (Hendrycks et al., 2021) and GSM8K (Cobbe et al., 2021). The Math-Shepherd and RLHFlow/Deepseek-PRM-Data datasets are synthetically generated following the rollout method proposed by Wang et al. (2024b). Similarly, the MATH-APS dataset is produced using the synthetic generation technique introduced by Luo et al. (2024). PRM800K, in contrast, consists of manually annotated labels and was specifically curated for the study by Lightman et al. (2024).

All PRMs are trained using the base LLMs of comparable model size and class, including Mistral-7B (Jiang et al., 2023), Llama-3.1-8B-Instruct (Dubey et al., 2024), and Qwen-2.5-Math 8B (Yang et al., 2024a).

*Table 3.* Training details of various Math PRMs

| PRM | Base Model | Training Data | Training Method |
|---|---|---|---|
| Math-PSA | Qwen-2.5-Math-7B-Instruct | PRM800K, Math-Shepherd and MATH-APS | LoRA |
| Math-Shepherd | Mistral-7B | Math-Shepherd | Full fine-tuning |
| Qwen-2.5-Math-PRM | Qwen-2.5-Math-7B-Instruct | PRM800K | Full fine-tuning |
| RLHFLow-Deepseek | Llama3.1-8B-Instruct | RLHFlow/Deepseek-PRM-Data | Full fine-tuning |
| LlamaPRM800K | Llama3.1-8B-Instruct | PRM800K | Full fine-tuning |
| QwenPRM800K | Qwen-2.5-Math-7B-Instruct | PRM800K | Full fine-tuning |

## D.2. Details of PRM Training

For training, we extract logits from the tokens + and − in the final layer of the LLM. The logit for + corresponds to a correct reasoning step, while the logit for − represents an incorrect step. We use four newline characters \n\n\n\n as the classification token, which is appended to the end of each reasoning step. We use standard cross-entropy loss and only compute it over our classification token.

For training our math PRMs on the PRM800K dataset (QwenPRM800K and LlamaPRM800K), we employ a batch size of 128 and perform full fine-tuning. For experiments on mixed-domain datasets, we reduce the batch size to 32 due to smaller dataset size.

All training is conducted over a single epoch. For full fine-tuning, we use a learning rate of $1.25 \times 10^{-6}$, while for LoRA-based fine-tuning, we use a learning rate of $1.0 \times 10^{-4}$.

# E. Additional PRM Comparisons

This appendix compiles a set of additional results that fill out the main paper's findings. Concretely, it is organized as follows:

- Table 4 is a more detailed version of Table 1 in the main text. It contains results over additional open source PRMs.

- Figure 18 provides side-by-side comparison of the effects of using three different aggregation strategies with WMV and BoN .

- Figure 19 shows WMV and BoN results when the generator is swapped from Llama-3.1-8B-Instruct to Llama-3.1-70B-Instruct

- Figure 20 shows results obtained when the PRM is downsized from Llama-3.1-8B-Instruct to a smaller model, Llama-3.1-3B-Instruct.

- Figures 21 and 22 provide WMNV and BoN over every MMLU-Pro category for VersaPRM.

- Figure 23 gives a domain-level comparisons of Beam Search and Monte-Carlo Tree Search when powered by VersaPRM versus a math-only PRM baseline.

- Figure 24 is a more detailed version of Figure 5—containing experimental results for additional tested domains.

## E.1. Evaluation Results for Math PRMs and VersaPRM Across all Categories

In this section, we list various other miscellaneous results

*Table 4.* Comparison among various math PRMs and VersaPRM on different domains in MMLU-Pro-CoT-Eval (Unlabeled) when using WMV with min-aggregation on $N = 16$ CoTs generated per question using Llama3.1-8B-Instruct. In parenthesis we report the relative difference between WMV and the MV baseline (WMV$-$MV). While WMV using math PRMs exhibit greater improvement in math and math-adjacent domains, there is no significant improvement on MV in other domains.

| Category | MV (Baseline) | Math-PSA | Math-Shepherd | Qwen-2.5-Math-PRM | RLHFLow-Deepseek | LlamaPRM800K | VersaPRM |
|---|---|---|---|---|---|---|---|
| All | 57.15 | 57.87(+0.72) | 57.66(+0.51) | 58.17(+1.02) | 57.59(+0.44) | 58.16(+1.01) | **61.22(+4.07)** |
| All except math | 56.61 | 56.82(+0.21) | 57.01(+0.40) | 57.32(+0.71) | 56.96(+0.35) | 57.71(+1.10) | **60.29(+3.68)** |
| Math | 62.40 | 64.20(+1.80) | 64.13(+1.73) | 67.20(+4.80) | 64.07(+1.67) | 65.40(+3.00) | **68.87(+6.47)** |
| Math-Adjacent | 56.75 | 57.98(+1.23) | 57.48(+0.73) | 58.30(+1.55) | 57.33(+0.58) | 58.27(+1.52) | **61.22(+4.47)** |
| Non-Math-Adjacent | 56.69 | 56.79(+0.10) | 57.14(+0.45) | 57.09(+0.40) | 57.02(+0.33) | 57.55(+0.86) | **60.00(+3.31)** |
| Chemistry | 58.67 | 60.47(+1.80) | 60.13(+1.46) | 60.67(+2.00) | 59.13(+0.46) | 60.47(+1.80) | **66.13(+7.46)** |
| Computer Science | 55.80 | 56.93(+1.13) | 56.07(+0.27) | 56.13(+0.33) | 56.07(+0.27) | 56.40(+0.60) | **58.60(+2.80)** |
| Engineering | 51.67 | 50.67(-1.00) | 51.07(-0.60) | 53.13(+1.46) | 51.87(+0.20) | 52.27(+0.60) | **55.27(+3.60)** |
| Physics | 58.53 | 61.87(+3.34) | 61.87(+3.34) | 61.47(+2.94) | 60.80(+2.27) | 61.47(+2.94) | **64.87(+6.34)** |
| Biology | 75.38 | 75.23(-0.15) | 75.38(+0.00) | 75.69(+0.31) | 75.77(+0.39) | 76.38(+1.00) | **80.00(+4.62)** |
| Health | 63.36 | 63.00(-0.36) | 63.93(+0.57) | 63.50(+0.14) | 63.57(+0.21) | 64.50(+1.14) | **65.50(+2.14)** |
| Psychology | 61.60 | 61.47(-0.13) | 61.47(-0.13) | 62.27(+0.67) | 61.47(-0.13) | 61.87(+0.27) | **64.53(+2.93)** |
| Business | 61.34 | 61.95(+0.61) | 62.21(+0.87) | 63.02(+1.68) | 62.21(+0.87) | 62.62(+1.28) | **64.50(+3.16)** |
| Economics | 62.00 | 62.67(+0.67) | 62.33(+0.33) | 62.53(+0.53) | 62.67(+0.67) | 62.40(+0.40) | **64.27(+2.27)** |
| Law | 35.93 | 35.72(-0.21) | 37.24(+1.31) | 36.28(+0.35) | 36.07(+0.14) | 36.90(+0.97) | **43.86(+7.93)** |
| History | 49.20 | 49.00(-0.20) | 49.87(+0.67) | 49.40(+0.20) | 49.40(+0.20) | 49.87(+0.67) | **50.67(+1.47)** |
| Philosophy | 44.83 | 44.90(+0.07) | 44.70(-0.13) | 45.17(+0.34) | 44.56(-0.27) | 45.30(+0.47) | **49.13(+4.30)** |
| Other | 55.53 | 55.80(+0.27) | 55.47(-0.06) | 56.07(+0.54) | 55.87(+0.34) | 57.07(+1.54) | **59.00(+3.47)** |

## E.2. WMV and BoN using different aggregation methods

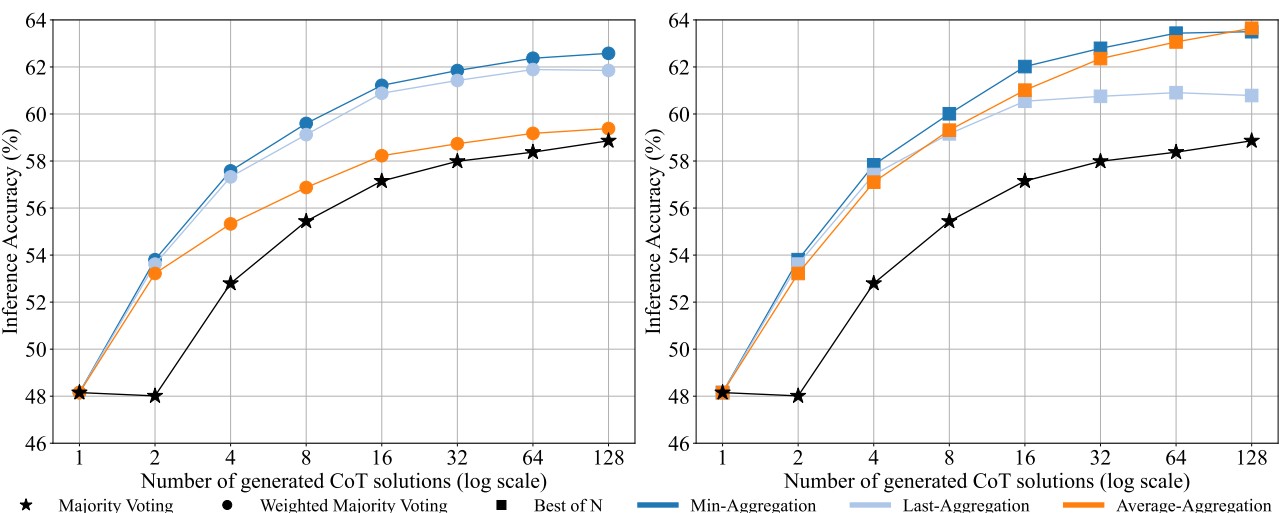

*Figure 18.* Comparison of WMV (left) and BoN (right) using VersaPRM with different reward aggregations on MMLU-Pro-CoT-Eval (Unlabeled). The CoTs are generated using Llama 3.1 8B Instruct. Overall, min-aggregation brings the largest inference performance boost.

## E.3. Larger Generator Inference with PRM Rewarding

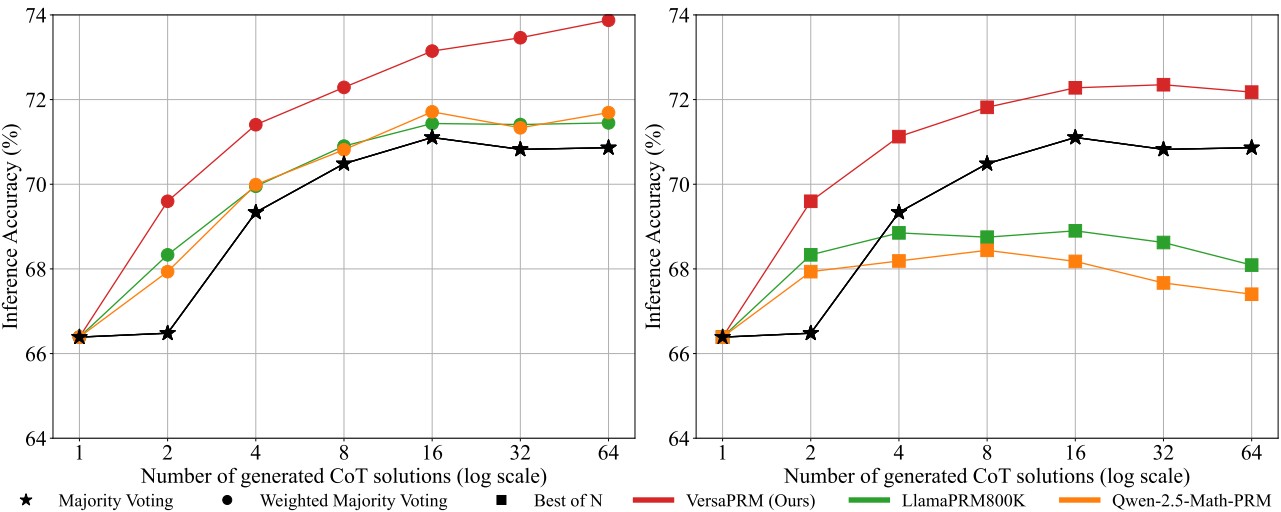

*Figure 19.* Comparison of WMV (left) and BoN (right) using VersaPRM against math PRMs on MMLU-Pro-CoT-Eval (Unlabeled). We use min-aggregation and the CoTs are generated using Llama-3.1-70B-Instruct. Similar trends to using 8B model as the generator are observed, indicating that our Multi-Domain PRM can generalize across generators with different capacities.

## E.4. Inference with PRM of smaller model size

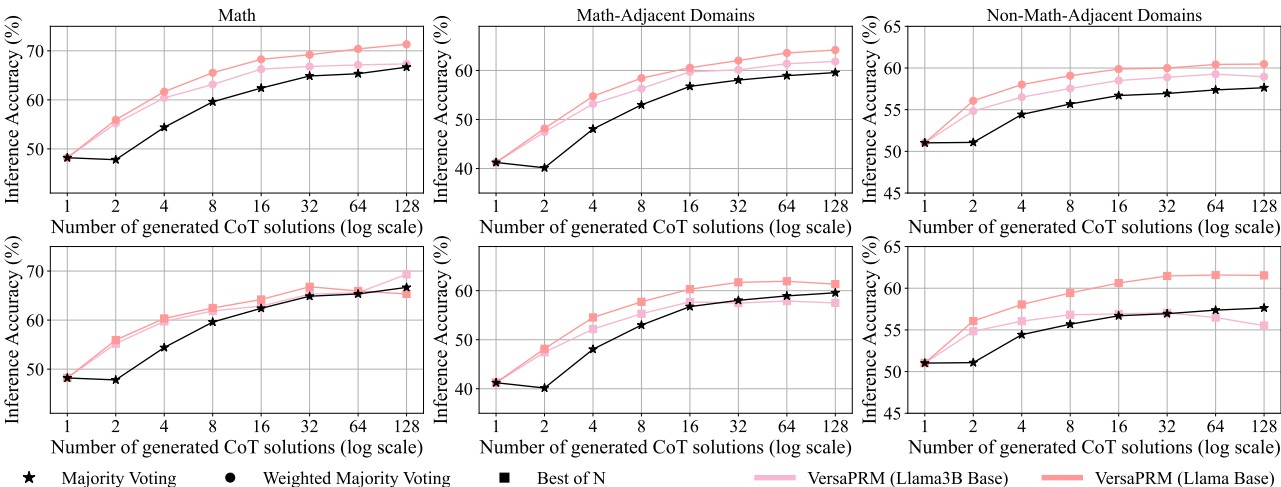

*Figure 20.* Comparison of WMV (top) and BoN (bottom) using VersaPRM (Llama3B Base), a PRM based on Llama-3.2-3B-Instruct and trained on our multi-domain dataset. We use min-aggregation and the CoTs are generated using Llama-3.1-8B-Instruct. Compared with using VersaPRM (Llama Base), which applies the same training data and configurations but is based on Llama-3.1-8B-Instruct, VersaPRM (Llama3B Base) brings a less significant performance boost. However, the overall trends are similar, indicating the efficacy of the inference pipeline using PRM.

## E.5. Comparison of VersaPRM against Other Open-Source Math PRMs on WMV and BoN by Category

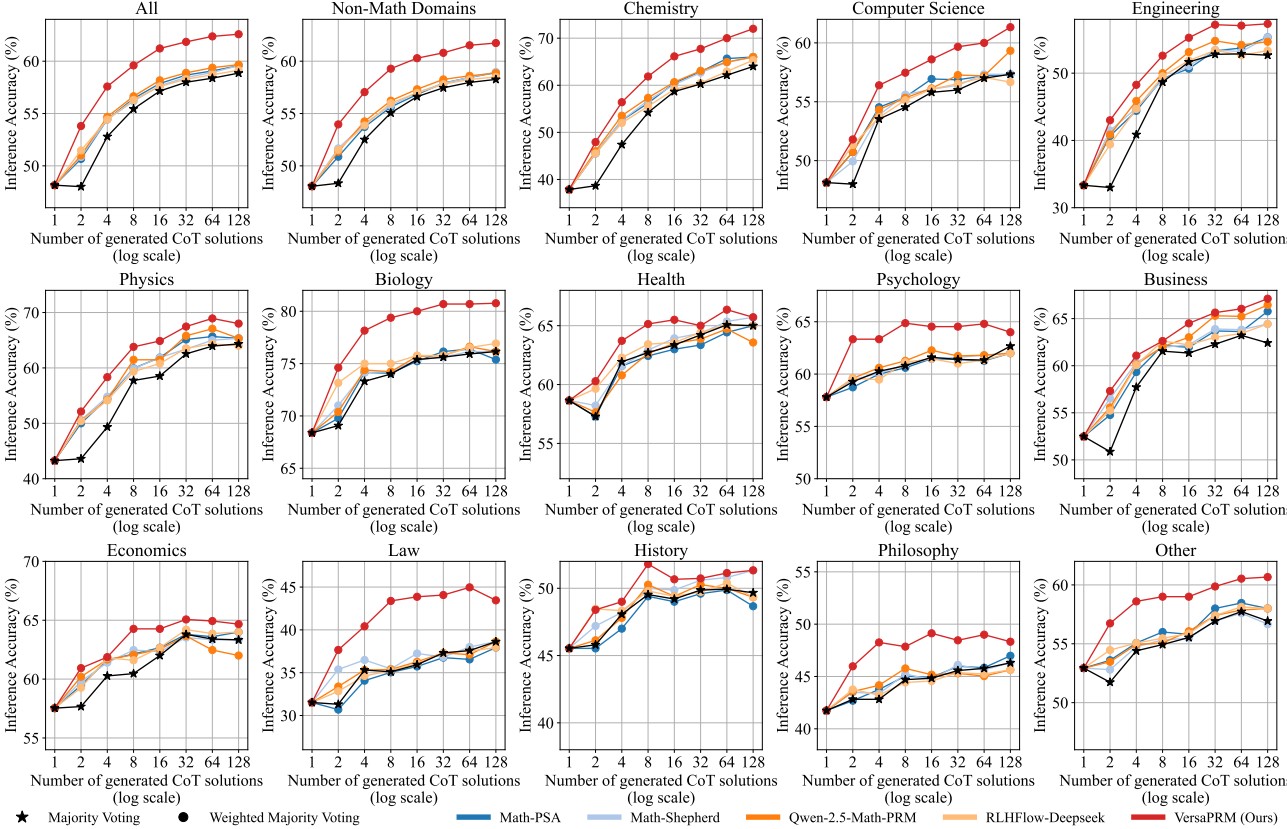

*Figure 21.* Comparison of WMV using VersaPRM against open-source PRMs on more other categories of MMLU-Pro-CoT-Eval (Unlabeled). We use min-aggregation and the CoTs are generated using Llama-3.1-8B-Instruct.

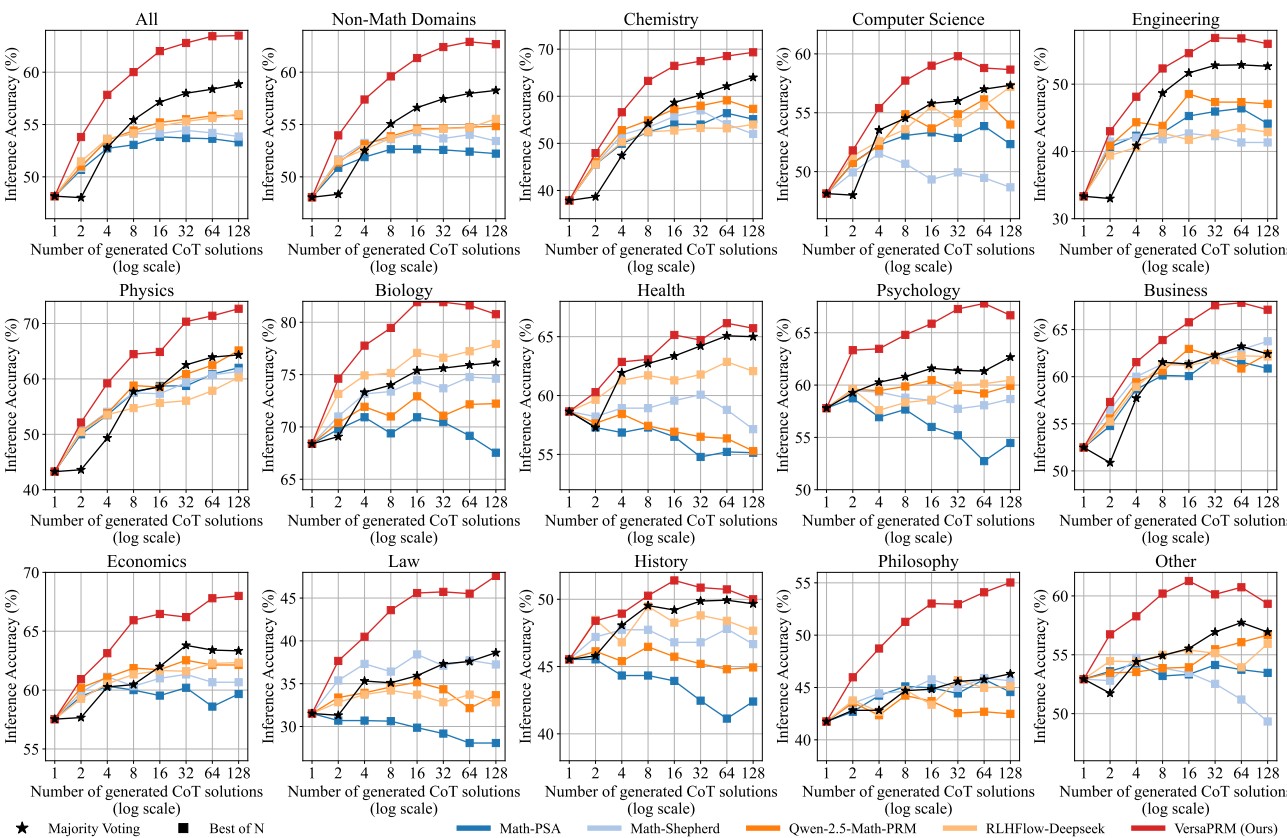

*Figure 22.* Comparison of BoN using VersaPRM against open-source PRMs on more other categories of MMLU-Pro-CoT-Eval (Unlabeled). We use min-aggregation and the CoTs are generated using Llama-3.1-8B-Instruct.

## E.6. Comparison of VersaPRM against Other Open-Source Math PRMs on MCTS and Beam Search by Category

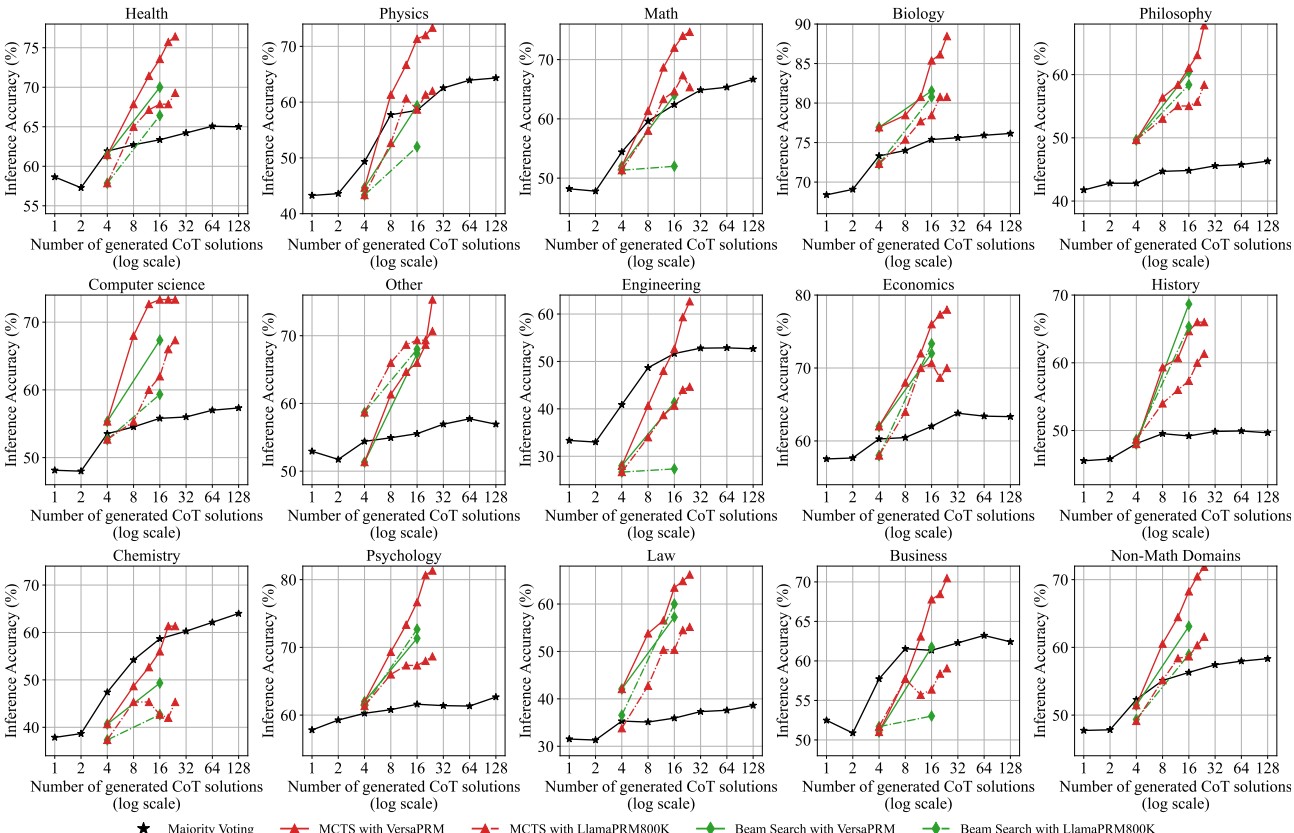

*Figure 23.* Comparison of VersaPRM and LlamaPRM800K with beam search and MCTS. In more other categories from MMLU-Pro-CoT-Eval (Unlabeled), VersaPRM achieves better performance.

## E.7. Comparison of VersaPRM against VersaPRM Trained with One Categories Held-out

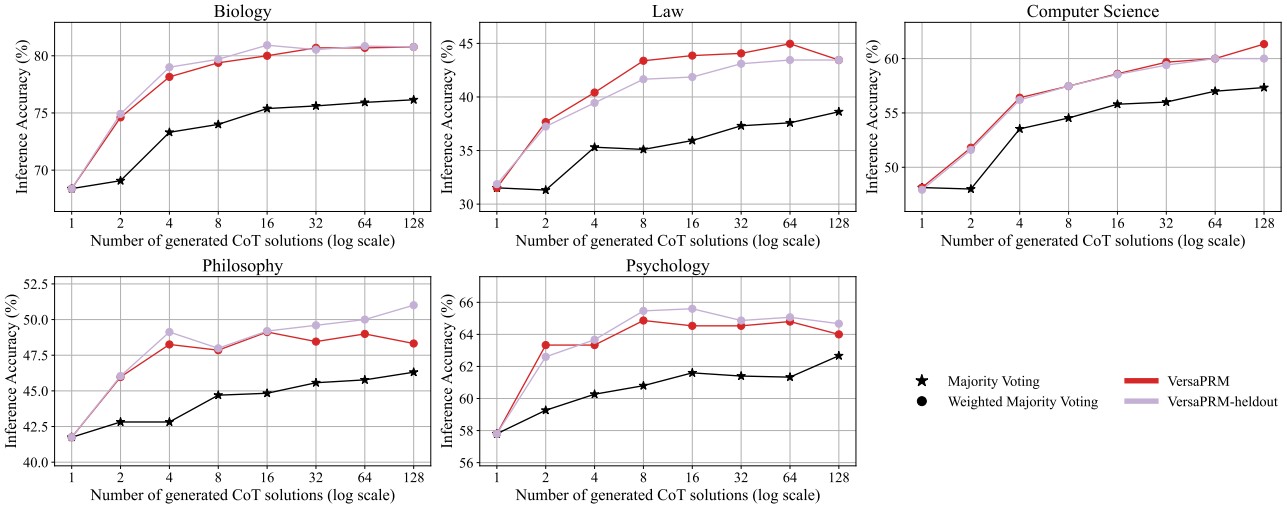

*Figure 24.* Hold-out domain ablation experiment results across all tested categories. WMV performance comparison of VersaPRM trained with each of the indicated domains held out versus the fully trained VersaPRM.

# F. Manual Analysis of VersaPRM and Math PRM Mistakes

In this section, we manually analyze a few handpicked examples of reward scores outputted by VersaPRM and a math PRM (specifically Qwen-PRM800K). For the following examples, steps **bolded and colored in red** denote steps manually inspected to be incorrect.

## F.1. Examples where the Math PRM Fails but VersaPRM Succeeds

**Question ID:** 1141_mmlu_law

**Category:** Law

---

**Question:**
A man and a woman were involved in a car accident that occurred when the man abruptly switched lanes in front of the woman without signaling. Unable to stop, the woman hit the man's car into a median. Immediately after the accident, as the two drivers emerged from their vehicles, a pedestrian, obviously emotional and upset from having just observed an accident, ran up to the man and shouted, "You careless driver, you. Didn't you ever learn that you're supposed to use a turn signal when you switch lanes?" The pedestrian is never identified and is not present at trial. The pedestrian's statement is...

**Options:**
**A.** inadmissible, because the bystander was not involved in the accident; **B.** admissible, because it is relevant to the case; **C.** admissible under the excited utterance exception; **D.** admissible as non-hearsay; **E.** admissible, even though it is hearsay; **F.** admissible under the present sense impression exception; **G.** inadmissible, because it is an opinion, not a fact; **H.** inadmissible, because the bystander cannot be identified and is not present at trial; **I.** inadmissible, because it is prejudicial to the defendant; **J.** admissible under the spontaneous statement exception.

**Answer:** C

---

**CoT 1:**

1. The pedestrian's statement is a spontaneous statement made while the person is under the stress of excitement caused by the event, and before they have time to reflect on the matter.
2. The statement is an out-of-court statement that is being offered for the truth of the matter asserted, which is that the man switched lanes without signaling.
3. Therefore, the statement is hearsay, but it is admissible under the excited utterance exception to the hearsay rule.
4. **The answer is (J).**

**Math PRM rewards:**
0.75, 0.82, 0.84, 0.86

**VersaPRM rewards:**
0.98, 0.95, 0.93, 0.17

---

**Explanation:**
The Math PRM does not check that the selected answer lines up with the prior reasoning steps.

---

**CoT 2:**

1. The statement is an outburst from the bystander immediately following the accident, and it describes what he just saw.
2. **This meets the requirements of the present sense impression exception, which allows for the admission of statements describing or explaining an event or condition made while the declarant is perceiving the event or condition.**

3. The statement is describing the man's actions in switching lanes without signaling, and it is being made immediately after the accident.

4. **The answer is (F).**

**Math PRM rewards:**
0.94, 0.92, 0.96, 0.87

**VersaPRM rewards:**
1.00, 0.43, 1.00, 0.09

---

**Explanation:**
The Math PRM fails to detect that the wrong hearsay exception was applied.

---

**CoT 3:**

1. The pedestrian's statement is an out-of-court statement that is being offered for the truth of the matter asserted; **I.**e., that the man was a careless driver who failed to use a turn signal.

2. The statement is an admission by a bystander, but it is still hearsay.

3. **There is no exception to the hearsay rule that allows a statement to be admitted if it was made by a bystander.**

4. **The answer is (E).**

**Math PRM rewards:**
0.70, 0.90, 0.77, 0.65

**VersaPRM rewards:**
0.97, 0.80, 0.42, 0.08

---

**Explanation:**
The Math PRM fails to recognize that there are exceptions to the hearsay rule such as the excited utterance exception. Thus, the PRM might have mistakenly judged the logical structure as correct without verifying if the statements were factually accurate in the legal context.

---

**Question ID:** 2518_mmlu_psychology

**Category:** Psychology

---

**Question:**
Which of the following terms is the best synonym for test validity?

**Options:**
**A.** Dependability; **B.** Authenticity; **C.** Generality; **D.** Reliability; **E.** Stability; **F.** Truthfulness; **G.** Precision; **H.** Consistency; **I.** Replicability

**Answer:** F

---

**CoT:**

1. Validity refers to how accurately a test measures what it is supposed to measure.

2. Reliability refers to how consistently a test measures what it is supposed to measure.

3. However, there is another term that is often confused with validity, which is reliability.

4. **The answer is (D).**

**Math PRM rewards:**
0.98, 0.99, 0.86, 0.80

**VersaPRM rewards:**
1.00, 1.00, 0.96, 0.02

---

**Explanation:**
The Math PRM again does not check that the selected answer lines up with the prior reasoning steps.

---

**Question ID:** 5362_mmlu_other

**Category:** Other

---

**Question:**

Which of the following is an example of devolution?

**Options**

**A.** Allowing states to decide how to meet federally mandated greenhouse gas emissions targets; **B.** The federal government setting a minimum wage that all states must meet; **C.** A state being prevented from implementing a fuel efficiency standard that is more stringent than that set by the federal government; **D.** A "race to the bottom" in which states compete with one another to have the fewest environmental regulations; **E.** A state being forced to comply with federal immigration policies; **F.** The federal government establishing nationwide educational standards; **G.** A state choosing to implement stricter gun control laws than those mandated by the federal government; **H.** A state that decriminalizes all drug use, contrary to federal law; **I.** A state that legalizes marijuana possession in conflict with federal law; **J.** The federal government overriding a state's decision to legalize same-sex marriage

**Answer:** A

---

**CoT:**

1. Devolution refers to the transfer of power or authority from a central government to a lower level of government, such as a state or local government.

2. **Option C states that a state is prevented from implementing a fuel efficiency standard that is more stringent than that set by the federal government. This is an example of devolution because the federal government is limiting the power of the state to act on its own, effectively "devolving" power back to the federal level.**

3. **The answer is (C).**

**Math PRM rewards:**
0.98, 0.63, 0.89

**VersaPRM rewards:**
1.00, 0.10, 0.47

---

**Explanation:**

The Math PRM likely lacks the domain-specific understanding of political science and government structure. It does not recognize that devolution inherently involves granting more autonomy rather than restricting it.

---

## F.2. Examples where VersaPRM Fails

**Question ID:** 4410_mmlu_chemistry

**Category:** Chemistry

**Question:**
Question: The solubility product constant of magnesium hydroxide is $8.9 \times 10^{-12}$, calculate its solubility in (a) water and (b) .05M NaOH.

**Options:**
**A.** $1.4 \times 10^{-4} mol/liter$ and $3.7 \times 10^{-9} mol/liter$; **B.** $1.5 \times 10^{-4} mol/liter$ and $3.8 \times 10^{-9} mol/liter$; **C.** $1.1 \times 10^{-4} mol/liter$ and $3.0 \times 10^{-9} mol/liter$; **D.** $1.2 \times 10^{-4} mol/liter$ and $3.5 \times 10^{-9} mol/liter$; **E.** $1.8 \times 10^{-4} mol/liter$ and $3.2 \times 10^{-9} mol/liter$; **F.** $2.0 \times 10^{-4} mol/liter$ and $2.5 \times 10^{-9} mol/liter$; **G.** $1.6 \times 10^{-4} mol/liter$ and $4.0 \times 10^{-9} mol/liter$; **H.** $2.2 \times 10^{-4} mol/liter$ and $3.9 \times 10^{-9} mol/liter$; **I.** $9.0 \times 10^{-5} mol/liter$ and $4.5 \times 10^{-9} mol/liter$; **J.** $1.3 \times 10^{-4} mol/liter$ and $3.6 \times 10^{-9} mol/liter$

**Answer: J**

**CoT 1:**

1. Let's calculate the solubility of magnesium hydroxide in water and in 0.05M NaOH.

2. The solubility product constant ($K_{sp}$) is given by:

3. $K_{sp} = [Mg^{2+}][OH^-]^2$.

4. For magnesium hydroxide, the dissolution reaction is:

5. $Mg(OH)_2(s) \rightleftharpoons Mg^{2+}(aq) + 2OH^-(aq)$.

6. Let x be the solubility of magnesium hydroxide in moles per liter (M).

7. Then, $[Mg^{2+}] = x$ and $[OH^-] = 2x$.

8. Substituting these values into the $K_{sp}$ expression, we get:

9. $8.9 \times 10^{-12} = x(2x)^2$
   $8.9 \times 10^{-12} = 4x^3$
   $x^3 = 2.225 \times 10^{-12}$
   $x = 1.17 \times 10^{-4} \approx 1.2 \times 10^{-4} M$.

10. Therefore, the solubility of magnesium hydroxide in water is approximately $1.2 \times 10^{-4} M$.

11. When 0.05M NaOH is added, the concentration of $OH^-$ ions increases, and the equilibrium shifts to the left. The new concentration of $OH^-$ ions is $0.05 + 2x$.

12. However, the concentration of $Mg^{2+}$ ions is still x, because the NaOH does not react with Mg2+.

13. Substituting these values into the $K_{sp}$ expression, we get:

14. $8.9 \times 10^{-12} = x(0.05 + 2x)^2$

15. This equation is difficult to solve algebraically, but we can make an approximation. The concentration of $OH^-$ ions is much larger than the concentration of $Mg^{2+}$ ions, so we can assume that $0.05 + 2x \approx 0.05$.

16. Substituting this value into the $K_{sp}$ expression, we get:

17. $8.9 \times 10^{-12} \approx x(0.05)^2$
    $x \approx 3.56 \times 10^{-9} M$

18. Therefore, the solubility of magnesium hydroxide in 0.05M NaOH is approximately $3.6 \times 10^{-9} M$.

19. The answer is (J).

**Math PRM rewards:**

0.99, 0.98, 0.98, 0.97, 0.99, 0.99, 0.97, 0.96, 0.87, 0.98, 0.82, 0.84, 0.92, 0.95, 0.83, 0.91, 0.68, 0.87, 0.91

**VersaPRM rewards:**

1.00, 1.00, 1.00, 1.00, 1.00, 1.00, 1.00, 1.00, 0.88, 1.00, 0.82, 0.93, 1.00, 0.80, 0.28, 0.99, 0.72, 0.97, 0.43

---

**Explanation:**

Neither the math PRM or VersaPRM label the calculations in step 9 as incorrect. VersaPRM marks step 15 as incorrect.

---

**CoT 2:**

1. **The solubility product constant is given by $K_{sp} = [Mg^{2+}][OH^-]$.**

2. The solubility of magnesium hydroxide is $s$.

3. **We know that $K_{sp} = s^2 K_w$, since $s$ moles of magnesium hydroxide will produce $s$ moles of $Mg^{2+}$ and $2s$ moles of $OH^-$, and we know that $K_w = 10^{-14}$.**

4. We can now calculate the solubility of magnesium hydroxide:

5. $s = \sqrt{\frac{K_{sp}}{K_w}} = \sqrt{\frac{8.9 \times 10^{-12}}{10^{-14}}} = \sqrt{8.9 \times 10^2} = 9.43 \times 10^{-2} M.$

6. In the presence of .05M NaOH, we have an excess of hydroxide ions, so we can assume that the concentration of $OH^-$ is approximately equal to .05M.

7. We can now calculate the solubility of magnesium hydroxide:

8. $K_{sp} = [Mg^{2+}][OH^-] = [Mg^{2+}](.05).$

9. $\Rightarrow [Mg^{2+}] = \frac{K_{sp}}{.05} = \frac{8.9 \times 10^{-12}}{.05} = 1.78 \times 10^{-10} M.$

10. The answer is (J).

**Math PRM rewards:**

0.97, 0.99, 0.41, 0.94, 0.40, 0.87, 0.95, 0.98, 0.89, 0.63

**VersaPRM rewards:**

1.00, 1.00, 0.53, 1.00, 0.69, 0.96, 1.00, 0.95, 0.83, 0.12

---

**Explanation:**

While the math PRM is able to detect the errors in steps 3 and 5, neither PRM recognizes the incorrect formula in step 1, which leads to the incorrect result in step 9. This suggests that both PRMs have gaps in their knowledge about this domain.

---

**Question ID:** 8925_mmlu_math

**Category:** Math

---

**Question:**
Question: In Canada in 2014, the average wholesale price of soybeans was $0.24 per pound. In 2015, the average wholesale price of soybeans was $0.16 per pound. If a retailer purchased 20,000 pounds of soybeans in 2014 and in 2015, what was the percent change in the retailer's expenses from 2014 to 2015?

**Options:**
**A.** -25%; **B.** 33%; **C.** -33%; **D.** -8%; **E.** 16%; **F.** 8%; **G.** -20%; **H.** -50%; **I.** 25%; **J.** 0%

**Answer:** C

---

**CoT:**

1. The retailer purchased 20,000 pounds of soybeans in 2014 and 2015.

2. The cost of 20,000 pounds of soybeans in 2014 was $0.24 * 20,000 = 4800$.

3. The cost of 20,000 pounds of soybeans in 2015 was $0.16 * 20,000 = 3200$.

4. **The percent change in the retailer's expenses from 2014 to 2015 is $\frac{4800-3200}{4800} = \frac{1600}{4800} = \frac{1}{3} = 33\%$.**

5. **The answer is (B).**

**Math PRM rewards:**
0.99, 0.98, 0.99, 0.18, 0.20

**VersaPRM rewards:**
1.00, 1.00, 1.00, 0.68, 0.78

**Explanation:**
VersaPRM does not recognize the error in the order of subtraction in step 4, which causes the result to have an incorrect sign.

**Question ID:** 862_mmlu_business

**Category:** Business

---

**Question:**

Question: The following entries appeared in the ledgers of Kane's Record Store: Beginning Inventory, Purchases, Ending Inventory = $16,000, $58,000, $14,000. Find the store's rate of inventory turnover.

**Options:**

**A.** 4 times; **B.** 6 times; **C.** 2 times; **D.** 3 times; **E.** 4.5 times; **F.** 8 times; **G.** 3.5 times; **H.** 7 times; **I.** 5 times; **J.** 2.5 times

**Answer:** A

---

**CoT:**

1. "To find the rate of inventory turnover, we need to use the following formula:

2. Inventory Turnover = Cost of Goods Sold / Average Inventory.

3. **The cost of goods sold is the difference between purchases and ending inventory:**

4. **Cost of Goods Sold = Purchases - Ending Inventory**
   **= $58,000 - $14,000**
   **= $44,000.**

5. The average inventory is the average of beginning and ending inventory:

6. Average Inventory = (Beginning Inventory + Ending Inventory) / 2
   = ($16,000 + $14,000) / 2
   = $15,000.

7. Now, we can calculate the inventory turnover:

8. **Inventory Turnover = Cost of Goods Sold / Average Inventory**
   **= $44,000 / $15,000**
   **= 2.93 (or approximately 3 times).**

9. **The answer is (D).**

**Math PRM rewards:**

0.97, 0.98, 0.56, 0.55, 0.97, 0.93, 0.99, 0.21, 0.23

**VersaPRM rewards:**

1.00, 0.99, 0.96, 0.93, 1.00, 1.00, 1.00, 0.99, 0.95

---

**Explanation:**

VersaPRM does not recognize the improper method of calculating the cost of goods sold in step 4, which causes it to accept incorrect results.

# G. Other Experiments

## G.1. Evaluation on Open-Ended Law Questions

We run an additional (preliminary) experiment testing the ability of Versa-PRM to work on open-ended problems. Specifically, we first take the 145 law questions from MMLU-Pro-CoT-Eval and filter out the questions that contain the wording "which of the following [X] is correct", so that we have a total of 116 questions remaining.

For each question, similar to before, we generate 16 responses to each question using Llama-3.1-8B-Instruct. However, differently, we *remove* all the answer choices from the question in the prompt. Thus the question effectively becomes open ended. The specific prompt is given in Figure 26.

We next use BoN to rerank these CoTs using both VersaPRM and a math PRM (Qwen-PRM800K) as the reward model. We grade the final selected CoTs for correctness using Llama-3.3-70B-Instruct (Figure 27). The final results are seen in Figure 25. We see that with VersaPRM, accuracy increases with larger values of $N$—albeit it saturates around $N = 6$. On the other hand the math PRM fails entirely and in fact sees a negative change in accuracy with increasing values of $N$. This thus provides preliminary evidence that VersaPRM can also work in open-ended non-multiple choice settings.

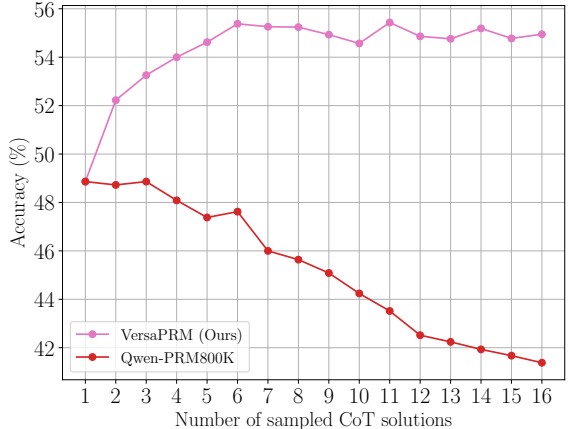

*Figure 25.* Performance of VersaPRM vs. a math PRM (Qwen-PRM800K) on 116 open-ended law questions from MMLU-Pro-CoT-Eval (i.e., by removing the answer choices from the original multiple choice questions), using BoN (min-aggregation). VersaPRM improves with larger $N$, while the math PRM fails to generalize.

> Given the following question, respond with the best possible answer.
>
> `{open-ended question}`
>
> Your response should end with "Final Response: [your answer]", where [response to the question] should be replaced with your actual response. Each reasoning step should be separated by two newline characters.
> Let's think step by step.

*Figure 26.* Prompt used to generate the CoTs used for BoN evaluation. The generator model is Llama3.1-8B-Instruct. Crucially, we do not provide the answer choices for the question when inserting it into the prompt.

## G.2. Evaluation of Iterative Refinement Using PRM

In addition to the test-time scaling methods presented in the main body of the paper, another approach to solution generation is iterative refinement. In this method, the initial response generated by the LLM is critiqued by a critic model, whose feedback is then used by the LLM to iteratively refine its answer (Madaan et al., 2023; Xu et al., 2024; Xi et al., 2024).

> Given the following open-ended question, reference answer, and student response, evaluate the student's response for correctness.
>
> Question: {question}
> Reference Correct Response: {reference answer}
> Student Response: {parsed answer}
> Provide a step-by-step analysis of the student's response using the reference answer as a guide. Your response should end with [GOOD] if you believe the students response to be correct and valid with respect to the question and [BAD] otherwise.
> Let's think step by step.

*Figure 27.* Prompt used by the grader LLM (Llama3.3-70B-Instruct) to evaluate correctness of the CoTs selected via BoN

We evaluate iterative refinement with one iteration, specifically employing the PRM as the critic model. Initially, we score all CoT reasoning in the MMLU-Pro-CoT-Eval dataset using VersaPRM and Qwen-2.5-Math-PRM. We then prompt Llama-3.1-8B-Instruct to generate revised responses, following the template in Figure 28. After discarding responses that either timed out or failed parsing, our refined dataset comprises 220,640 CoT responses covering 1,754 questions from MMLU-Pro-CoT-Eval.

We compare the average Pass@1 rate—computed by calculating the proportion of CoTs with correct final answers per question, then averaging across all questions—for the original CoT dataset, refinement using VersaPRM, and refinement using Qwen-2.5-Math-PRM. These methods yield Pass@1 rates of $51\%$, $45\%$, and $44\%$, respectively.

These results indicate that VersaPRM slightly outperforms Qwen-2.5-Math-PRM, further confirming the results in the main paper that it has superior multi-domain generalization. However, notably, both refinement methods perform worse than the baseline without refinement. This outcome implies that the current refinement prompt in this experiment may not be optimal—potentially overly biasing the LLM by the provided scores—and suggests that additional prompt engineering is necessary for better performance. We leave a more detailed study of iterative refinement with PRMs to future work.

> Given the following question and answer choices:
>
> {question}
>
> Below is a previously generated chain-of-thought reasoning for this question. Each step in the reasoning has an associated score (between 0 and 1), provided by a reward model predicting the correctness of that step:
>
> {cot and step scores}
>
> Re-examine and reconsider each step carefully. As you rethink the reasoning process, explicitly use the provided scores as indicators to identify potentially correct or incorrect reasoning steps. Clarify your reasoning and adjust it accordingly based on these scores.
> Clearly separate each reasoning step by two newline characters.
> After thoroughly reconsidering the reasoning process, select the best answer choice.
> Conclude your response clearly with: "The answer is (X).", where X is the letter corresponding to your chosen answer.
> Let's think step by step.

*Figure 28.* Prompt used to generate refined response given feedback

Step 0: To determine the effect of the purchase on GDP, we need to consider the components of GDP.
Score 0: 0.9765625

Step 1: GDP (Gross Domestic Product) is calculated as the sum of Consumption (C), Investment (I), Government Spending (G), and Net Exports (X).
Score 1: 0.9453125

Step 2: In this case, XYZ Corporation is buying the Matisse painting for its board room. This is an example of Investment (I), which is the purchase of capital goods or assets that are expected to increase future production.
Score 2: 0.6953125

Step 3: Since the purchase is an investment, it will increase the Investment (I) component of GDP.
Score 3: 0.92578125

Step 4: The answer is (J).
Score 4: 0.93359375

*Figure 29.* Example of a CoT and corresponding step scores that would go in the {cot and step scores} part of the prompt in Figure 28

