# OpenReview forum: "VersaPRM: Multi-Domain Process Reward Model via Synthetic Reasoning Data"
_ICML.cc/2025/Conference — ICML 2025 oral_

### Official Review · Reviewer_EnQL · 2025-03-13

**Overall Recommendation:** 4

**Summary:**

The paper introduces VersaPRM, a multi-domain Process Reward Model (PRM) designed to improve reasoning abilities across diverse domains beyond mathematics. Traditional PRMs have been primarily trained on mathematical reasoning tasks and fail to generalize effectively to other disciplines such as Law, Philosophy, and Biology. To address this issue, the authors propose a synthetic data generation and annotation pipeline that produces step-wise labeled reasoning data across multiple domains. VersaPRM is trained on this synthetic multi-domain dataset, leading to improved generalization and performance across various test-time inference strategies (e.g., Weighted Majority Voting (WMV), Best-of-N (BoN), Beam Search, and Monte Carlo Tree Search (MCTS)). Empirical evaluations on MMLU-Pro-CoT-Eval demonstrate that VersaPRM outperforms existing math-focused PRMs in non-mathematical domains, with a notable 7.9% improvement in Law compared to a baseline.

**Claims And Evidence:**

Overall, the paper presents strong empirical support for its claims, but there are some areas where additional clarification could be beneficial.

1. PRMs trained only on mathematical data fail to generalize to other domains.
The results in Tables clearly demonstrate that existing math-trained PRMs (e.g., Math-Shepherd, Qwen-2.5-Math-PRM) perform poorly in non-math domains, supporting this claim.

2. VersaPRM improves reasoning performance across multiple domains through synthetic multi-domain training.
The authors provide multiple ablation studies showing that PRMs trained on diverse reasoning data consistently outperform math-trained PRMs across Law, Philosophy, and Biology.

3. The synthetic data generation process produces high-quality multi-domain reasoning labels.
This claim is not fully supported, as the precision is around 70-75%, lower than 80% claimed from OpenAI.

**Essential References Not Discussed:**

no

**Experimental Designs Or Analyses:**

Yes, the experimental design is robust, but a few areas could be strengthened.

Strengths:
1. The synthetic data generation pipeline is clearly described, using Llama-3.1-8B to generate Chain-of-Thought (CoT) reasoning and Llama-3.1-70B as an auto-labeler.
2. Multiple reranking methods (WMV, BoN) and search strategies (Beam Search, MCTS) ensure that results are not limited to a single inference-time approach.
3. The ablation studies confirm that performance gains come from multi-domain data rather than additional training data volume.

Potential Weaknesses:
1. The impact of synthetic data noise is not fully explored. Although manual evaluation suggests 75% accuracy, additional breakdowns on how mislabels affect model performance would be beneficial.
2. The choice of backbone LLMs (Llama vs. Qwen) is not fully explored—for example, the effect of using even larger models (e.g., GPT-4 or DeepSeek-R1) remains an open question.

**Methods And Evaluation Criteria:**

The proposed methodology and evaluation criteria are aligned with the research goals.

1. Benchmark Datasets:
The authors evaluate VersaPRM on MMLU-Pro-CoT-Eval, which covers 14 diverse domains (Math, Physics, Law, Biology, Philosophy, etc.), ensuring a comprehensive evaluation.

2. Evaluation Metrics:
The use of WMV, BoN, and search-based methods (MCTS, Beam Search) effectively captures both accuracy and inference-time performance.

3. Baselines & Comparisons:
The comparisons with four open-source math PRMs (Math-Shepherd, Qwen-2.5-Math-PRM, etc.) establish a strong benchmark.

**Other Comments Or Suggestions:**

There are few typo and words repeated. Please check carefully.

**Other Strengths And Weaknesses:**

Weakness
1. Auto-Labeling Reliability
The auto-labeling method for reasoning steps, while cost-effective, introduces potential noise. The manual evaluation (75% accuracy) suggests that a portion of the training data may contain incorrect annotations, which could impact PRM reliability.

Strength
1.Novelty & Relevance
The paper addresses an important gap in PRM research by extending process supervision beyond mathematical domains.
The synthetic data generation approach, which includes automated reasoning step labeling, is an innovative method to scale PRM training.
Empirical Contributions

2. The study provides thorough comparisons between math-focused PRMs and VersaPRM, showing clear performance improvements across multiple domains.
The ablation studies and test-time inference analyses validate the generalization capacity of VersaPRM.

3.Reproducibility & Open Science
The authors commit to open-sourcing all datasets, model checkpoints, and code, facilitating reproducibility and further research.

4.Strong Experimental Setup
The evaluation includes rigorous baselines, including open-source math PRMs and multiple test-time inference techniques (e.g., majority voting, beam search, Monte Carlo Tree Search).
The inclusion of large-scale reasoning models such as DeepSeek-R1 strengthens the credibility of the findings.

**Questions For Authors:**

no

**Relation To Broader Scientific Literature:**

The paper builds upon and extends prior research in reward modeling and reasoning in LLMs, particularly in the following areas:

Process Reward Models (PRMs) vs. Outcome Reward Models (ORMs):
1. Prior works (e.g., Uesato et al., 2022; Lightman et al., 2024) established that PRMs outperform ORMs in math reasoning.
2. VersaPRM expands PRM utility beyond math, demonstrating improved performance in law, biology, and philosophy.

Test-Time Compute and Self-Improvement Methods:
1. The paper aligns with test-time inference strategies like Tree of Thoughts (ToT) (Yao et al., 2024) but enhances them with PRM-guided reranking.
2. Demonstrates that process reward models remain useful even for strong reasoning models (e.g., DeepSeek-R1).

Synthetic Data for PRM Training:
1. Prior works (e.g., Wang et al., 2024; Zheng et al., 2024) used synthetic process data for math reasoning.
2. VersaPRM generalizes this approach to non-math domains, demonstrating broader applicability.

**Theoretical Claims:**

The paper does not include formal theoretical proofs but provides empirical justifications for its findings.

---

> ### Author Rebuttal · Authors · 2025-04-01
>
> We thank the reviewer for your meaningful feedback and recognizing that (i) our paper presents strong empirical results and robust experiments, (ii) it addresses an important gap in PRM research, (iii) it provides thorough comparison between math PRMs and VersaPRM, and (iv) it facilitates open science with open-sourcing.
>
> > The synthetic data generation process produces high-quality multi-domain reasoning labels. This claim is not fully supported, as the precision is around 70-75%, lower than 80% claimed from OpenAI.
>
> Our synthetic labeling achieves ~90% of the accuracy of human-labeled PRM800K (the target benchmark), despite requiring no manual effort. The slightly lower precision (70-75%) reflects the cost-effectiveness trade-off: OpenAI’s 80% required expensive human annotation, while our method is fully automated.
>
> > The impact of synthetic data noise is not fully explored. Although manual evaluation suggests 75% accuracy, additional breakdowns on how mislabels affect model performance would be beneficial.
> > Auto-Labeling Reliability The auto-labeling method for reasoning steps, while cost-effective, introduces potential noise. The manual evaluation (75% accuracy) suggests that a portion of the training data may contain incorrect annotations, which could impact PRM reliability.
>
> We did not attempt to further decrease the noise in the training dataset, as we found 75% label accuracy sufficient for VersaPRM to generalize robustly across domains and inference methods.
> From a preliminary analysis, we observe that many mislabels stem from a failure to identify faulty reasoning due to incorrect factual information.
>
> Further noise reduction (e.g., via stronger labeling models) is promising but deferred to future work.
>
> > The choice of backbone LLMs (Llama vs. Qwen) is not fully explored—for example, the effect of using even larger models (e.g., GPT-4 or DeepSeek-R1) remains an open question.
> > Impact of larger-scale LLMs on PRM performance (e.g., GPT-4 vs. Llama-3).
>
> In Section 6.4, we provide initial results with DeepSeek-R1 (700B parameters) in the law domain. In addition, we are currently testing this approach in the biology domain and will include these results. Broader exploration of large models for PRM initialization/auto-labeling remains future work.
>
> > More examples of mislabeled CoTs to analyze failure cases.
>
> We provide examples CoTs mislabeled by math PRM but not by VersaPRM in [this link](https://drive.google.com/file/d/1IWVXEgDCpdhDHZ_zixL3ZHtbhq9xWGQq/view?usp=sharing). We will include more diverse examples of errors that VersaPRM also makes.
>
> > There are few typo and words repeated. Please check carefully.
>
> Thank you for noting this—we will thoroughly proofread the manuscript.
>
> **Final note:** Thank you again for the comments and we appreciate the thorough review!

---

### Official Review · Reviewer_pcYc · 2025-03-13

**Overall Recommendation:** 3

**Summary:**

This paper introduces VersaPRM, a multi-domain Process Reward Model (PRM) designed to enhance reasoning capabilities across diverse domains beyond mathematics. The authors identify that current PRMs are predominantly trained on mathematical data and demonstrate poor generalization to non-mathematical domains. To address this limitation, they develop a synthetic data generation and annotation pipeline that uses Llama-3.1-8B-Instruct to generate Chain-of-Thought (CoT) reasoning steps and Llama-3.1-70B-Instruct to automatically label these steps. Using this synthetic data, they train VersaPRM, which shows performance gains across multiple domains of MMLU-Pro compared to math PRMs. The authors contribute their multi-domain PRM, the MMLU-Pro-CoT-Train (Labeled) dataset, and open-source their implementation.

**Claims And Evidence:**

While the paper makes several reasonable claims, there are some gaps in the evidence presented. The core claim that VersaPRM generalizes better across domains (VersaPRM outperforms math PRMs) is supported by comparative evaluations, but a fundamental question remains unanswered: Is the improvement primarily from the distillation of a stronger model's knowledge (Llama-70B) into a smaller model, rather than from the PRM architecture or training methodology itself? The authors do not adequately control for this variable.

**Essential References Not Discussed:**

The paper covers parts of the relevant prior work comprehensively (areas covered in related work section). However, the paper lacks discussion of several important related areas:

1. Knowledge distillation literature, which is essentially what the authors are doing when using Llama-70B to generate training data for a smaller model.
2. Literature on model alignment through AI feedback (e.g., Constitutional AI approaches), which uses similar techniques of having larger models provide feedback to smaller models.
3. Recent work comparing the cost-effectiveness of using larger models at test time versus training specialized models, which would provide important context for evaluating their approach.
4. Work on zero-shot evaluation capabilities of large models, which would be relevant for understanding the baseline capability of Llama-70B as a direct evaluator.

**Experimental Designs Or Analyses:**

The experimental design is comprehensive and sound:

1. The authors use appropriate baselines, including multiple open-source math PRMs and majority voting.

2. The evaluation across all domains of MMLU-Pro with consistent metrics allows for fair comparison.

3. The experiments with DeepSeek-R1 (Figure 7) provide preliminary evidence (limited to the law subset) that the approach benefits even stronger reasoning models.


However, Several issues exist in the experimental design:

1. Lack of a proper baseline using Llama-70B directly as an evaluator at test time, which would help isolate whether the benefit comes from distilling Llama-70B's knowledge or from the PRM training approach.

2. The experiments do not adequately assess the cost-effectiveness of the approach. Using a 70B model to generate training data is expensive, and it's unclear if this cost is justified by the performance gains.

3. The validation of auto-labeling quality uses a small sample size (30 questions), which could be expanded for more robust validation of the dataset quality.

**Methods And Evaluation Criteria:**

The evaluation using MMLU-Pro is appropriate and provides a solid testbed with reasoning problems across 14 domains.

The comparison across multiple test-time computation methods (MV, WMV, BoN, Beam Search, MCTS) shows the robustness of their approach across different inference-time techniques. However, authors do not adequately control for computational costs across these methods, making efficiency comparisons difficult.

The evaluation of auto-labeling quality through manual inspection provides a reasonable sanity check, though a larger sample could strengthen confidence in the dataset quality. For a dataset spanning multiple domains with thousands of examples, 30 examples is insufficient to establish confidence in labeling quality.

However, what's fundamentally unclear is whether the paper's approach is simply knowledge distillation from Llama-70B to a smaller model. The authors do not include a crucial baseline: using Llama-70B directly as an evaluator at test time. This would help determine whether training a separate PRM provides any advantage over directly using the stronger model's judgments.

**Other Comments Or Suggestions:**

- A more detailed error analysis showing specific examples where VersaPRM succeeds but math PRMs fail would enhance understanding of the model's strengths
- Visualizing what constitutes "good" versus "bad" reasoning steps across different domains could provide interesting insights
- Exploring more efficient inference techniques that require fewer candidate solutions would enhance practical applicability
- Investigating whether the approach could be extended to open-ended generation tasks beyond multiple-choice questions would be valuable

**Other Strengths And Weaknesses:**

**Strengths:**
- The problem addressed is significant and practical, as it enables more effective use of test-time computation across diverse domains
- The synthetic data generation pipeline is effective
- The comprehensive ablations provide valuable insights about what factors matter for multi-domain PRM effectiveness
- The open-sourcing of data, code and models will benefit the research community

**Weaknesses:**
- Fundamental ambiguity about whether improvements come from knowledge distillation or PRM training
- The manual evaluation of auto-labeling quality uses a relatively small sample
- Limited discussion of potential biases inherited from the generator and labeler LLMs
- The improvements in some domains (e.g., History) remain modest compared to other domains, but reasons for this variability aren't deeply analyzed
- No ablation study about the synthetic data generation pipeline. This appears to be the main contribution of the paper but the process is not dissected well.

**Questions For Authors:**

1. Have you evaluated using Llama-70B directly as an evaluator at test time compared to your VersaPRM? This would help isolate whether the benefit comes from distilling Llama-70B's knowledge or from your specific PRM training methodology.

2. Have you analyzed the specific types of reasoning errors that VersaPRM is better at identifying compared to math PRMs across different domains? This could help explain the varying degrees of improvement observed across domains.

3. Have you explored using VersaPRM for iterative refinement of reasoning (rather than just reranking), where feedback from the PRM guides the generation process itself? This could potentially lead to higher quality reasoning with fewer total generations.

**Relation To Broader Scientific Literature:**

This work relates to:

1. It extends work on Process Reward Models (Lightman et al., 2024; Wang et al., 2024b; Luo et al., 2024) by addressing their domain generalization limitations.

2. It builds on test-time computation techniques for LLM reasoning (Snell et al., 2024; Yao et al., 2024; Wan et al., 2024) by proposing a practical method for constructing efficient PRM evaluators.

**Theoretical Claims:**

The paper does not make significant theoretical claims requiring proof verification. The formal definitions provided in Section 3 for PRMs and aggregation methods are straightforward and align with established concepts in the literature.

---

> ### Author Rebuttal · Authors · 2025-04-01
>
> We thank the reviewer for the thoughtful feedback. We've addressed your concerns below.
>
> > Multiple test-time computation methods (MV, WMV, BoN, Beam Search, MCTS) but computational costs across these methods are not controlled
>
> We clarify the computational costs of test-time methods: WMV and BoN scale with the number of generated CoT solutions $N$. Beam Search’s number of beams $B$ is equivalent to $N$, and MCTS scales with the branching factor times the number of iterations. Figure 6 compares MCTS and Beam Search in terms of computational cost for an equivalent number of generated CoT solutions.
>
> However, our goal is not to compare inference-time methods against each other, but to show that VersaPRM consistently outperforms math PRMs across all these methods.
>
> > 30 examples is insufficient to establish confidence in labeling quality
>
> We expanded our sample to 60 questions. The revised analysis shows that 78% of responses marked correct by the autolabeler were accurate (95% CI: 0.68–0.89), and 70% of those marked incorrect were accurate (95% CI: 0.62–0.78). Based on this, we estimate 74% of CoT responses in the training set are correctly labeled. We will label more examples for the revision.
>
> >  Using Llama-70B directly as an evaluator at test time compared to VersaPRM
>
> Thank you for your valuable suggestion. We provide an [additional experiment](https://drive.google.com/file/d/1rwev0yBT0scwyMj1Tgc6nXaUGy1EsFad/view?usp=sharing) comparing VersaPRM against Llama-70B directly as a judge on MMLU-CoT-Eval. VersaPRM outperforms Llama-70B across all values of $N$.
>
> Recall the training data for VersaPRM is labeled by Llama-70B with access to the correct answers during prompting, ensuring high-quality step-wise labels (Section 5.3). At inference, however, Llama-70B lacks prior knowledge of the correct answer, limiting its judging effectiveness as a judge. Thus, the gains stem from our PRM methodology, not merely Llama-70B’s knowledge.
>
> > The experiments do not adequately assess the cost-effectiveness of the approach; Using a 70B model to generate training data is expensive
>
> We used Llama-70B via AWS Bedrock batch inference to label ~85K CoTs in our training data, costing ~$50. This is far cheaper than manual labeling or stepwise rollouts. We will clarify this in the final paper.
>
> > Lack discussion of several important related areas
>
> Thank you for suggesting potential discussion on important related areas. We will add in detailed discussion of these areas in the final paper.
> > Limited discussion of potential biases inherited from the generator and labeler LLMs
>
> Thank you for raising this issue. Since both generator and labeler LLMs are Llama models, VersaPRM may perform better on Llama-based inference. A detailed study of the bias would be important, especially for alignment. We will add this discussion in the revision.
>
> > The reason behind moderate improvements in some domains (e.g., History) compared to other domains
>
> Manual inspection of questions in domains with moderate improvements (e.g., history and health) reveals that these primarily test factual recall over reasoning. We posit that aggregation methods like BoN and WMV are less effective in these domains, as success depends largely on the LLM’s factual knowledge–not reasoning quality–and whether the PRM can recognize the correct fact. Therefore, we suspect PRMs struggle in these domains due to weaker pretrained knowledge of health/history.
>
> > No ablation study about the synthetic data generation pipeline.
>
> Section 5.3 presents an ablation on auto-labeling prompts (Lines 275–282). We also tested removing ground-truth answers, which dropped agreement rates from 83% to 60% for CoT originally autolabeled as correct and 70% to 40% for those autolabeled as incorrect. We are conducting further end-to-end PRM evaluations using ablated training data and will update results accordingly.
>
> > A more detailed error analysis showing specific examples where VersaPRM succeeds but math PRMs fail; Visualizing what constitutes "good" versus "bad" reasoning steps across different domains
>
> We provide mislabeled CoTs from math PRMs that VersaPRM correctly labels [here](https://drive.google.com/file/d/1IWVXEgDCpdhDHZ_zixL3ZHtbhq9xWGQq/view?usp=sharing). We will include examples of VersaPRM errors.
>
> > VersaPRM for iterative refinement of reasoning (rather than just reranking)
>
> Section 6.3 tests MCTS and Beam Search, which achieve strong results with significantly fewer computations.
>
> > Open-ended generation tasks extension
>
> We include an [experiment on open-ended law questions](https://drive.google.com/file/d/1wYJO1daWMX-hngTnNg0hnrXX79Ggnhvt/view?usp=sharing).
>
> **Final Note**: We appreciate the in-depth feedback and hope our comments and additional experiments address the questions.  If our responses have addressed your concerns, please kindly consider raising the score.

---

> > ### Comment · Reviewer_pcYc · 2025-04-04
> >
> > Hi,
> >
> > Thanks for the detailed reply. I appreciate
> > 1/ Experiment on Llama-70B directly as an evaluator at test time compared to VersaPRM
> > 2/ expanded labelling analysis to 60 questions which makes their claim more grounded
> > 3/  providing the additional discussions based on my comments.
> > 4/ Expaning their study to open ended law questions.
> >
> > I think some of my comments were not clear enough, mainly
> >
> > > VersaPRM for iterative refinement of reasoning (rather than just reranking)
> > > > Section 6.3 tests MCTS and Beam Search, which achieve strong results with significantly fewer computations.
> >
> > What I meant here is that the authors should consider a process where VersaPRM is used tp flag appropriate and inappropriate answers with both fed back to inference LLM to refine its answer. Not only using VersaPRM to select the best one. I understand this might be out of scope though.
> >
> > > No ablation study about the synthetic data generation pipeline.
> > >> Section 5.3 presents an ablation on auto-labeling prompts (Lines 275–282). We also tested removing ground-truth answers, which dropped agreement rates from 83% to 60% for CoT originally autolabeled as correct and 70% to 40% for those autolabeled as incorrect. We are conducting further end-to-end PRM evaluations using ablated training data and will update results accordingly.
> >
> > This is definitely a good starting point but I think that more can be done in this direction. There is quite rich literature on automated labelling such as [https://arxiv.org/abs/2308.08998, https://arxiv.org/abs/2408.04614] and this angle is not explored well.
> >
> > ** With that in mind, I think the additions the authors promised to add to the paper merit a score increase**.

---

> > > ### Author Response · Authors · 2025-04-08
> > >
> > > Thank you for your added clarification and for recognizing our new experiments. We will ensure all promised additions are included in the final paper.
> > >
> > > > What I meant here is that the authors should consider a process where VersaPRM is used to flag appropriate and inappropriate answers with both fed back to inference LLM to refine its answer. Not only using VersaPRM to select the best one. I understand this might be out of scope though.
> > >
> > > We appreciate the clarification and agree that using VersaPRM not just for selection but also as feedback to refine the LLM's answer is an interesting idea. Prior work has explored iterative refinement using natural language critiques or scalar/binary scores over entire responses as the form of external feedback. We find this idea promising and will include an experiment in the final version of the paper that tests this.
> > >
> > > Specifically, after the LLM generates an initial response, we will provide it with the step-level correctness scores assigned by the PRM and prompt it to revise its answer based on that feedback. We will conduct this experiment using both VersaPRM and baseline math PRMs to evaluate if VersaPRM leads to more effective refinement.
> > > > This is definitely a good starting point but I think that more can be done in this direction. There is quite rich literature on automated labelling such as [https://arxiv.org/abs/2308.08998, https://arxiv.org/abs/2408.04614] and this angle is not explored well.
> > >
> > > Thank you for highlighting these additional references. We agree that approaches like response-rewriting and self-training could further enhance synthetic data generation. Thus, in addition to our initial studies with counterfactual augmentation (Appendix A.3) which we ablated due to limited gains for the final trained PRM, we will extend our experiments to include the methods you suggested.
> > >
> > > In particular, we will experiment with 1) augmenting the training CoT data by using an LLM to generate alternate phrasings of existing steps that preserve their original meaning and 2) applying self-training by labeling additional CoT examples directly by VersaPRM, heuristically filtering them for correctness, and using them to further train VersaPRM. We will report the results of these experiments in the updated version of the paper.
> > >
> > > ---
> > >
> > > Thank you again for your time and constructive engagement—we’re grateful for your support in strengthening the paper!

---

### Official Review · Reviewer_baxV · 2025-03-14

**Overall Recommendation:** 4

**Summary:**

Process Reward Models (PRMs) have been effective in improving mathematical reasoning for Large Language Models (LLMs) by utilizing increased inference-time computation. However, their generalizability to non-mathematical domains remains unproven. This work demonstrates that current PRMs perform poorly in non-mathematical domains. To address this, VersaPRM is introduced—a multi-domain PRM trained on synthetic reasoning data generated through a novel data generation and annotation method. VersaPRM achieves consistent performance gains across diverse domains. For example, in the MMLU-Pro Law category, VersaPRM improves performance by 7.9% using weighted majority voting, significantly outperforming Qwen2.5-Math-PRM's 1.3% gain. Additionally, all data, code, and models for VersaPRM are open-sourced for community use.

**Claims And Evidence:**

Yes

**Essential References Not Discussed:**

N/A

**Experimental Designs Or Analyses:**

Although a comparative analysis with a variety of open source mathematical PRMs was conducted, the comparative analysis was mainly based on existing models and datasets. For some more advanced models or methods that have not yet appeared, the comparative advantages of the model may need to be re-evaluated.

[Exploring Mathematical Extrapolation of Large Language Models with Synthetic Data](https://aclanthology.org/2024.findings-acl.55/) (Li et al., Findings 2024)

**Methods And Evaluation Criteria:**

- Advantages
1. A new method called VersaPRM (Multi-domain Process Reward Model) is proposed. By training on synthetic reasoning data, the reasoning ability of large language models in non-mathematical fields is effectively improved, filling the gap in the application of existing process reward models (PRMs) in multiple fields.
2. A novel synthetic data generation and automatic annotation method is designed. The chain of reasoning (CoT) is generated and automatically annotated using LLM, which avoids the high cost and low efficiency of manual annotation, while ensuring data quality and diversity, providing rich materials for model training.
Comprehensive evaluation criteria: A variety of evaluation methods are used, including weighted majority voting (WMV), best N selection (BoN), beam search (Beam Search) and Monte Carlo tree search (MCTS), etc., to verify the performance of the model from different angles, which can fully reflect the performance of the model in different fields and different reasoning strategies.
3. All data, code and models are open source, which facilitates subsequent research and promotes further exploration and application of this field in academia and industry.
- Disadvantages
1. Although the automatic labeling method is used, its accuracy still has room for improvement (about 75%), and there may be some incorrectly labeled data, which may interfere with model training and affect the further improvement of model performance.
2. Although a variety of evaluation methods are used, these methods are mainly based on existing reasoning strategies and data sets, and may not fully cover all possible reasoning scenarios and fields. For some more complex or challenging reasoning tasks, the performance of the model may need further verification.

**Other Comments Or Suggestions:**

N/A

**Other Strengths And Weaknesses:**

N/A

**Questions For Authors:**

N/A

**Relation To Broader Scientific Literature:**

N/A

**Theoretical Claims:**

N/A

---

> ### Author Rebuttal · Authors · 2025-04-01
>
> We thank the reviewer for the feedback, and for acknowledging that (i) our method fills a gap in the existing literature on PRMs, (ii) we provide comprehensive evaluation criteria for our model, and (iii) our open-sourced code facilitates future exploration of this field.
>
> > Although the automatic labeling method is used, its accuracy still has room for improvement (about 75%), and there may be some incorrectly labeled data, which may interfere with model training and affect the further improvement of model performance.
>
> While the automatic labeling method introduces some noise (~75% accuracy), our PRM remains robust to moderate noise levels. This is evident from VersaPRM’s strong performance gains over previous math PRMs, despite being trained on noisy data. In this paper, we used Llama-70B due to budget constraints and as it was sufficient to create a model that outperforms previous math PRMs. That said, to further increase labeling accuracy, we can employ stronger models, such as GPT-4o, or reasoning models such as DeepSeek-R1 or OpenAI o1. This would give us higher accuracy labeling. Given time constraints for this response, we will aim to include an improved model with more accurately labeled data in the final version of the paper.
>
> > Although a variety of evaluation methods are used, these methods are mainly based on existing reasoning strategies and data sets, and may not fully cover all possible reasoning scenarios and fields. For some more complex or challenging reasoning tasks, the performance of the model may need further verification.
>
> We have evaluated VersaPRM across diverse test-time methods. We acknowledge the importance of broader validation and will discuss and explore additional reasoning scenarios in the final version.
>
> > Although a comparative analysis with a variety of open source mathematical PRMs was conducted, the comparative analysis was mainly based on existing models and datasets. For some more advanced models or methods that have not yet appeared, the comparative advantages of the model may need to be re-evaluated. Exploring Mathematical Extrapolation of Large Language Models with Synthetic Data (Li et al., Findings 2024)
>
> We will cite  “Exploring Mathematical Extrapolation of Large Language Models with Synthetic Data” in the revised version. This paper introduces a novel arithmetical puzzle task and demonstrates how fine-tuning on large-scale synthetic examples enables precise multi-step mathematical reasoning. The fine-tuned model not only solves the puzzles but also generalizes to harder, out-of-distribution problems involving larger numbers and the composing components of the arithmetical puzzle problem. While this approach is effective for math, we note its applicability to non-math domains remains unclear, as defining comparable structured tasks in those areas presents challenges.
>
> **Final note:** Thank you again for the comments. If you have any remaining questions, please do not hesitate to let us know.

---

> > ### Comment · Reviewer_baxV · 2025-04-02
> >
> > I think your rebuttal is more effective. In the future, you can try to introduce your research into chess[1] and other research directions. It may be an interesting attempt.
> >
> > All in all I think this is a good approach.
> >
> > I've changed the rating.
> >
> > Good luck
> >
> > [1] Zhang, Y., Han, X., Li, H., Chen, K., & Lin, S. (2025). Complete Chess Games Enable LLM Become A Chess Master. arXiv preprint arXiv:2501.17186.

---

> > > ### Author Response · Authors · 2025-04-08
> > >
> > > Thank you for the updated review! We will mention even more challenging domains like game playing in the discussion section of the final paper.

---

### Official Review · Reviewer_gKUV · 2025-03-16

**Overall Recommendation:** 4

**Summary:**

This paper describes an automated pipeline for annotating chain-of-thought rationales with stepwise correctness labels. The authors generate a set of these annotated CoTs for MMLU-Pro, then train a model to predict these correctness labels. The result is VersaPRM, a process reward model for reasoning beyond math domains. The authors evaluate VersaPRM as a reranker against several math-specific PRMs using weighted majority voting, as well as plain majority voting without a reward model; they additionally evaluate VersaPRM as a scoring function for MCTS and beam search. Results on held-out MMLU-Pro questions show that weighted majority voting with math-specific PRMs fails to improve on simple majority voting in non-math-adjacent domains, whereas WMV with VersaPRM provides clear improvements in all considered domains. The authors also demonstrate that VersaPRM works well as a scoring function for search, providing better accuracy scaling with inference budget than majority voting or search with a math-specific PRM.

**Claims And Evidence:**

The authors' claims about their method are basically well-supported - see "Experimental Designs or Analyses" for a nitpick.

**Essential References Not Discussed:**

I am not aware of any essential references that the authors missed (with the exception of work currently in submission). "To CoT or Not to CoT" (Sprague et al. '24) could help contextualize the authors' finding that math PRMs are ineffective outside math domains - and the authors' demonstration that WMV with VersaPRM outperforms MV on non-math domains is extra surprising (positively) in light of the conclusion from that work that CoT primarily supports math performance and not other domains.

**Experimental Designs Or Analyses:**

The comparisons the authors carry out in their main experiments are valid. However, to truly demonstrate that VersaPRM is "domain-general" I would have preferred to see some evaluation beyond just MMLU-Pro, especially since the training traces are gathered from it.

**Methods And Evaluation Criteria:**

MMLU-Pro makes sense as a benchmark to evaluate this approach.

**Other Comments Or Suggestions:**

On line 306 there is a missing space between "PRM" and "via".

**Other Strengths And Weaknesses:**

I'm glad to see folks working to broaden the applicability of CoT and test-time search beyond math, these are encouraging results.
The MCTS performance under VersaPRM is especially cool - looks like performance doesn't saturate with increasing budget nearly as quickly as it does with the other setups.

**Questions For Authors:**

N/A

**Relation To Broader Scientific Literature:**

There have been a handful of previous process reward modeling efforts in the literature, but as the authors note, these have mostly focused on the math domain, as this is where most step-by-step reasoning with LMs has been developed and evaluated due to easy answer verification. This is the first work I have seen attempt to train a domain-general reasoning PRM, but I may have missed one.

**Theoretical Claims:**

N/A

---

> ### Author Rebuttal · Authors · 2025-03-28
>
> We thank the reviewer for the thoughtful feedback on the paper, recognizing that (i) its claims are well supported and (ii) this work is the first attempt to broaden PRMs beyond the math domain. We will incorporate your suggested revisions into our final paper.
>
> > The comparisons the authors carry out in their main experiments are valid. However, to truly demonstrate that VersaPRM is “domain-general” I would have preferred to see some evaluation beyond just MMLU-Pro, especially since the training traces are gathered from it.
>
> While we do not currently have results for VersaPRM on datasets beyond MMLU-Pro, we have conducted additional hold-one-out evaluations, which we will include in the final paper. Specifically, we exclude one domain category (law, biology, or CS) from training and assess whether the resulting PRM generalizes to the held-out domain.
>
> Results are available [here](https://drive.google.com/drive/folders/1n1bMkNSQ9u923b9P_IVW1nl3ENmSJCec?usp=sharing). As shown, the WMV generalization performance of VersaPRM *trained with one domain held out* is comparable to that of the full model. This suggests that its generalization ability is not merely due to broader coverage of the training data, but rather indicates genuine domain-general reasoning capabilities.
>
> > I am not aware of any essential references that the authors missed (with the exception of work currently in submission). "To CoT or Not to CoT" (Sprague et al. '24) could help contextualize the authors' finding that math PRMs are ineffective outside math domains - and the authors' demonstration that WMV with VersaPRM outperforms MV on non-math domains is extra surprising (positively) in light of the conclusion from that work that CoT primarily supports math performance and not other domains.
>
> We appreciate this reference and will include it in Section 2. The following sentence that will be added:
>
> *“According to the work of Sprague et al. (2024), most of the reported advantages using CoT stem from math or math-related tasks.”*
>
> > I'm glad to see folks working to broaden the applicability of CoT and test-time search beyond math, these are encouraging results. The MCTS performance under VersaPRM is especially cool - looks like performance doesn't saturate with increasing budget nearly as quickly as it does with the other setups.
>
> Thank you for this positive feedback. We will emphasize it in the final version.
>
> > On line 306 there is a missing space between "PRM" and "via".
>
> Thanks for catching this–we will fix it.
>
> **Final Note:** We appreciate the reviewer’s insights and suggestions and are encouraged by the positive assessment of our work on broadening test-time scaling methods to domains beyond math.

---

> > ### Comment · Reviewer_gKUV · 2025-04-05
> >
> > I like the hold-one-out evaluation idea - that's a great experiment to include. Thanks for addressing my comments!

---

> > > ### Author Response · Authors · 2025-04-08
> > >
> > > Thank you for the follow-up. We are glad you like the hold-one-out evaluation, and appreciate the thoughtful feedback!

---

### Decision · Program_Chairs · 2025-05-01

**Decision:**

Accept (oral)

**Comment:**

In this work, the authors propose VersaPRM, a novel Process Reward Model (PRM) that is trained on synthetically generated data beyond math word problems. The paper includes a few key contributions:
1. the authors first demonstrate that the current PRMs perform poorly in non-mathematical domains (because they are trained exclusively on math word problems);
2. the authors propose a novel synthetic data generation pipeline to generate and label reasoning chains from MMLU-Pro tasks (multi-domain). Notebly, they use a smaller LLM (8b) to generate the reasoning chains, and use a larger LLM (70b) to label.
3. the authors train a multi-domain PRM using the above data, and show that fine-tuning with synthetic multi-domain data provides PRMs better generalizability.
4. the authors promise to open-source data, code and models.

The authors and reviewers had a productive discussion. Below are some of the salient discussion points:
1. Reviewers had concern that since the reasoning chains are collected from MMLU-Pro, the current evaluation may provide insufficient evidence on VersaPRM's generalizability (because test tasks are in-domain). The authors acknowledge that time does not allow to run experiments on new datasets. However, they proposed a new setting which exludes one domain category from training and assess whether the resulting PRM could generalize to that held-out domain. Results show that the resulting PRM is still comparable to the full model shown in the paper, this suggests the generalization comes from domain-generl reasoning capabilities rather than having tasks in-domain.
2. Reviewers had concern that the size of the data subset used to  assess the quality of the auto-labeled data was small (30 data points). The authors double the size and show the same results.
3. The authors provide additional experiment results that compares VersaPRM against using Llama70b directly as a judge. Results show that because the lack of ground-truth labels, directly using Llama70b as a judge does not provide as good performance. This weakens the assumption that VersaPRM's good performance may due to the distillation from stronger model (70b), and suggests that the performance gain is really from the PRM methodology.
4. The authors and reviewers discussed the potential of using even stronger models in data labelling, which could potentially further improve the data quality and thus the performance of the fine-tuned PRM.

The authors did a great job addressing the reviewers' concerns. In fact, the authors have successfully convinced two reviewers to increase their scores, this is rare in my batch.

Additional AC comments:
Thanks for integrating R1 experiments and analysis so fast! I acknowledge that R1 was released one week before the ICML deadline.

Overall, this is a strong paper with clear contribution to the ICML community. The fact that the authors promise to open-source everything (data, model, code) makes the work even valuable to researchers and practitoners in the sub-field of LLM reasoning, LLM-as-a-judge and verifier training. I recommend to accept this paper.